# MaskRIS: Semantic Distortion-aware Data Augmentation for Referring Image Segmentation

**Minhyun Lee**[*][†]                                                      *mh315.lee@samsung.com*
*AI Center, Samsung Electronics*

**Seungho Lee**[*]                                                         *sh622.lee@samsung.com*
*AI Center, Samsung Electronics*

**Song Park**                                                             *song.pxxk@gmail.com*

**Dongyoon Han**                                                          *dongyoon.han@navercorp.com*
*NAVER AI Lab*

**Byeongho Heo**[‡]                                                        *bh.heo@navercorp.com*
*NAVER AI Lab*

**Hyunjung Shim**[‡]                                                       *kateshim@kaist.ac.kr*
*Korea Advanced Institute of Science & Technology (KAIST)*

**Reviewed on OpenReview:** *https://openreview.net/forum?id=EtK4madHmc*

## Abstract

Referring Image Segmentation (RIS) is an advanced vision-language task that involves identifying and segmenting objects within an image as described by free-form text descriptions. While previous studies focused on aligning visual and language features, exploring training techniques, such as data augmentation, remains underexplored. In this work, we explore effective data augmentation for RIS and propose a novel training framework called Masked Referring Image Segmentation (MaskRIS). We observe that the conventional image augmentations fall short of RIS, leading to performance degradation, while simple random masking significantly enhances the performance of RIS. MaskRIS uses both image and text masking, followed by Distortion-aware Contextual Learning (DCL) to fully exploit the benefits of the masking strategy. This approach can improve the model's robustness to occlusions, incomplete information, and various linguistic complexities, resulting in a significant performance improvement. Experiments demonstrate that MaskRIS can easily be applied to various RIS models, outperforming existing methods in both fully supervised and weakly supervised settings. Finally, MaskRIS achieves new state-of-the-art performance on RefCOCO, RefCOCO+, and RefCOCOg datasets. Code is available at https://github.com/naver-ai/maskris.

## 1 Introduction

Referring Image Segmentation (RIS) (Hu et al., 2016) involves precisely delineating objects within an image based on text descriptions. Unlike semantic and instance segmentation, which are constrained to pre-defined classes, RIS offers unique flexibility by segmenting objects specified by free-form expressions. This capability has led to extensive applications in diverse domains, such as language-driven human-robot interaction (Shah et al., 2023) and advanced image editing (Chen et al., 2018; Patashnik et al., 2021). One of the main

---

[*]Equal contribution. [†]Work done during an internship at NAVER AI Lab.
[‡]Corresponding authors.

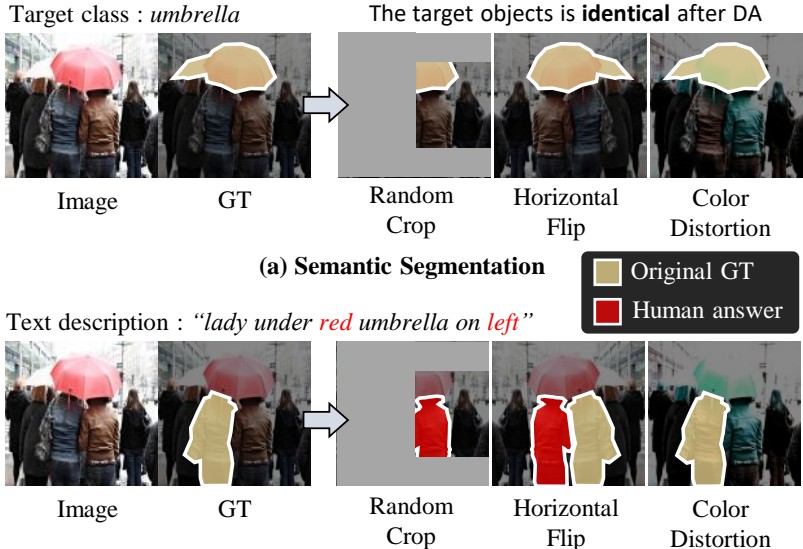

Figure 1: Conventional data augmentations (DA) in semantic segmentation are incompatible with referring image segmentation. Random crop and horizontal flip could change the referred object (*e.g.*, *"lady under the red umbrella on left"*) to another one, and color distortion could make the described object disappear.

challenges in RIS is effectively bridging the modality gap between visual and language features. Recent research (Liu et al., 2023c; Yang et al., 2022; Wang et al., 2022) has focused on developing sophisticated architectures for cross-modal alignment.

While significant progress has been made in aligning different modalities, effective training techniques for RIS remain underexplored. The trend toward developing larger and more complex models requires challenging training settings like data augmentation. Previous studies (Cubuk et al., 2018; 2020; Touvron et al., 2021) have demonstrated the crucial role of data augmentation in model training. However, designing data augmentation for RIS is challenging: standard augmentation used in semantic segmentation is often incompatible with RIS, as some textual descriptions directly conflict with augmentation.

As illustrated in Figure 1, spatial augmentation, such as random crop and horizontal flip, alters the meaning of "left", causing the target mask to become incorrect after augmentation. Similarly, color distortion also conflicts with color-specific descriptions like "red". Due to the incompatibilities, naïvely applying augmentations is not beneficial - our empirical analysis reveals that conventional data augmentation degrades RIS (Liu et al., 2023c) performance (Figure 3(a)). This finding explains why previous RIS studies (Kim et al., 2022; Wang et al., 2022; Yang et al., 2022; Liu et al., 2023c; Wu et al., 2023; Zhang et al., 2022) have taken a cautious approach to augmentation, predominantly relying on simple resizing while avoiding complex techniques that could severely distort the data.

In this paper, we explore an effective data augmentation framework to improve RIS model training. We propose a straightforward yet robust baseline, coined Masked Referring Image Segmentation (MaskRIS), which consists of two key components: (1) input masking as a data augmentation technique for RIS and (2) a Distortion-aware Contextual Learning (DCL) designed to maximize the benefits of input masking. In RIS tasks, it is crucial to preserve essential spatial information (*e.g.*, relative positions, ordering) and key attribute details (*e.g.*, color) for accurately understanding and segmenting objects based on referring expressions. Unlike conventional data augmentation, which often distorts these critical aspects, input masking minimizes such distortions by preserving spatial and attribute information while significantly expanding data diversity. Our approach extends input masking to both images and text to strengthen the model's capability to handle visual and linguistic complexity. Masking parts of the text encourages the model to infer missing

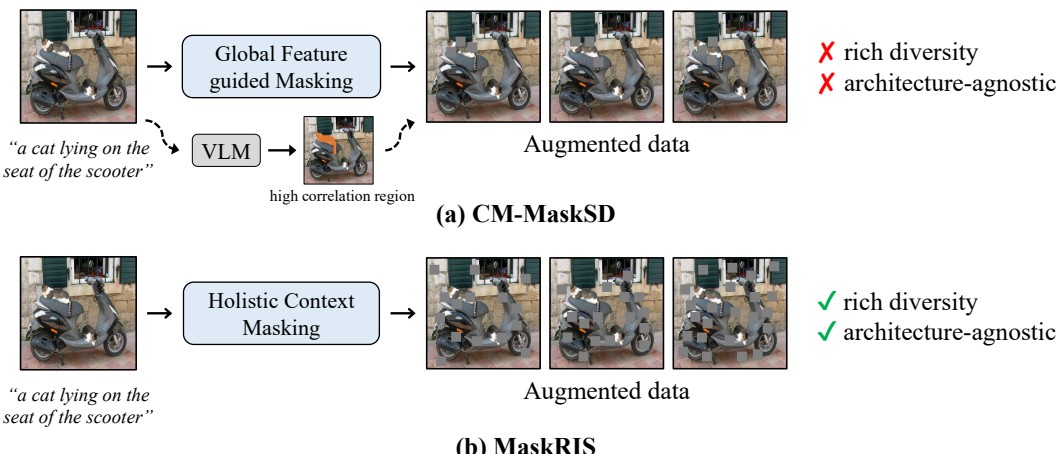

Figure 2: Comparison with CM-MaskSD (Wang et al., 2024). (a) CM-MaskSD employs global feature guided masking via a vision-language model (VLM), such as CLIP (Radford et al., 2021), which requires a pre-aligned vision–language representation. It focuses on high-correlation regions but lacks diversity and architecture-agnostic properties. (b) MaskRIS adopts holistic context masking, yielding richer diversity and improved architecture independence.

or ambiguous details, improving its understanding of diverse referring expressions and reducing reliance on specific terms. This masking strategy directly addresses the need for a robust RIS training framework, helping to overcome the performance bottleneck caused by limited augmentation.

To fully exploit the advantages of input masking, we propose a Distortion-aware Contextual Learning (DCL) framework that processes inputs through two complementary paths: a primary path with original inputs and a secondary path with masked inputs. The primary path ensures baseline training stability and maintains model accuracy, while the secondary path enhances data diversity and robustness. The distillation loss aligns predictions from both paths, encouraging the model to make consistent predictions on both original and masked inputs. This DCL framework enhances feature robustness to masked inputs through the secondary path, while the primary path provides stability and mitigates potential harmful effects of severe distortion. Additionally, DCL acts as a regularizer, adding beneficial challenges to the RIS task. The regularization effect aligns with findings from Mobahi et al. (2020), where self-distillation is shown to constrain model representations, reducing overfitting and enhancing robustness - a process similarly achieved by aligning original and masked predictions in our framework.

Finally, we note that although MaskRIS shares certain surface-level similarities with the concurrent work CM-MaskSD (Wang et al., 2024), it is conceptually and technically distinct. As illustrated in Figure 2, CM-MaskSD employs masking guided by cross-modal correlations, where a global text feature is compared against all image patch tokens (and vice versa for a global image feature with text tokens). This correlation-based masking selects the most text-relevant patches and distills them through the [CLS]-token representation, resulting in an object-centric, token-level self-distillation scheme. However, this design inherently relies on a pre-aligned vision–language representation, limiting its generalizability. In contrast, MaskRIS introduces holistic context masking across both image and text, which not only preserves spatial cues and avoids augmentation-induced semantic flips (*e.g.*, "left" and "right", color changes), but also yields richer diversity by exposing the model to a broader range of masked contexts. Our Distortion-aware Contextual Learning (DCL) aligns *segmentation outputs* from a clean primary path and a masked secondary path, enforcing pixel-wise consistency while maintaining stability and diversity. This enables reasoning under incomplete or ambiguous multi-modal cues; empirically, masking both modalities outperforms masking only one (see Sec. 4.3). Importantly, MaskRIS is free from pre-aligned cross-modal features, architecture-agnostic, and adds no parameters or inference overhead, thereby departing clearly from prior masking-based RIS methods.

Our contributions are as follows: (1) We provide a comprehensive analysis of data augmentation techniques for robust RIS training, identifying limitations in previous methods for the first time. (2) We propose a holistic context masking strategy that perturbs both image and text, increasing data diversity through varied,

yet coherent partial contexts while reducing semantic conflicts across modalities. (3) We introduce Distortion-aware Contextual Learning (DCL), a dual-path distillation framework that enforces pixel-level consistency between original and masked predictions. Unlike prior [CLS] token–dependent schemes tied to specific architectures, our approach combines holistic masking with DCL to enable stable and architecture-agnostic training. (4) Our method achieves significant performance gains over existing RIS methods, including weakly supervised approaches, establishing new state-of-the-art results on RefCOCO, RefCOCO+, and RefCOCOg datasets. Moreover, we show that the proposed framework also benefits Referring Expression Comprehension (REC), highlighting its broader applicability beyond RIS.

## 2 Related Work

**Referring Image Segmentation.** Referring Image Segmentation (RIS) aims to segment objects in images based on text descriptions. This field evolved from early CNN-RNN/LSTM combinations for image and text processing (Margffoy-Tuay et al., 2018; Shi et al., 2018; Hu et al., 2016; Li et al., 2018) to advanced approaches using cross-modal attention (Liu et al., 2017; 2021a; Ye et al., 2019; Hu et al., 2020; Hui et al., 2020; Ding et al., 2021) and pre-trained transformer encoders (Wang et al., 2022; Kim et al., 2022; Yang et al., 2022; Liu et al., 2023c). While previous studies have focused on image-text alignment, our study shifts focus to the training strategy in RIS, offering a complementary approach to enhance existing studies. This orthogonal perspective allows us to enjoy the performance benefits established by previous methods. Our concurrent work, NeMo (Ha et al., 2024), introduces a mosaic-based augmentation that combines a target image with carefully selected negative samples. While effective, it requires additional overhead for constructing and retrieving from a negative sample pool. In contrast, our approach provides a more efficient framework that enhances data diversity with a broader analysis of augmentation types and model robustness.

**Masked Input Training.** Masked input training is a self-supervised learning technique where random parts of input data are masked, and the model learns to predict these masked elements from visible data. Originally successful in natural language processing (NLP) as masked language modeling (MLM) (Devlin et al., 2018; Liu et al., 2019), it expanded to computer vision as masked image modeling (MIM) showing both training efficiency and strong generalization performance (Bao et al., 2021; Xie et al., 2022; He et al., 2022). These masking strategies are also employed to develop robust feature representations. RandomErasing (Zhong et al., 2020) and Cutout (DeVries & Taylor, 2017) erase randomly selected rectangle areas to reduce over-fitting and handle occlusion. MIC (Hoyer et al., 2023) leverages context clues from the target domain through masked image consistency for domain adaptation. SMKD (Lin et al., 2023) uses masked knowledge distillation for few-shot classification.

Recently, MagNet (Chng et al., 2024) applies masked text training to improve vision-language alignment in RIS, where masked words are predicted using image features. Similarly, CM-MaskSD (Wang et al., 2024) applies masked input training using object-centric masking and CLIP-based patch-text alignment to refine feature representations. Although both methods improve cross-modal learning, their main focus is on feature alignment. In contrast, MaskRIS is inspired by the semantic conflicts from data augmentation in RIS. To address this issue, we performed a systematic analysis of augmentation-induced inconsistencies and their impact on RIS performance. Based on these findings, MaskRIS introduces a holistic masking strategy and a distortion-aware training framework, explicitly designed to improve model robustness in RIS.

**Self-distillation.** Self-distillation leverages the model itself as both teacher and student, preventing the need for separate teacher networks to facilitate knowledge distillation (Hinton et al., 2015). PS-KD (Kim et al., 2021) demonstrates the utility of softening hard targets through teacher model predictions, acting as an effective regularization strategy. Some studies (Zhang et al., 2019; Phuong & Lampert, 2019) explored self-distillation by employing an entire network as the teacher and an early exit network as the student. CoSub (Touvron et al., 2023) improved model performance in visual recognition tasks with sub-model-based self-distillation. MaskSub (Heo et al., 2023) introduced a drop-based technique for sub-model self-distillation, improving model performance and cost efficiency for image classification tasks. To fully exploit the benefits of input masking, we employ self-distillation between the predictions of the original and masked input.

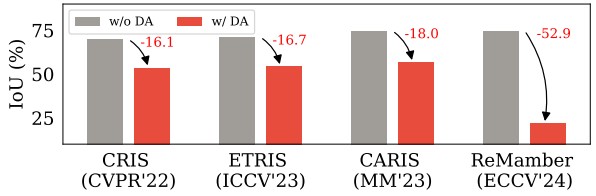 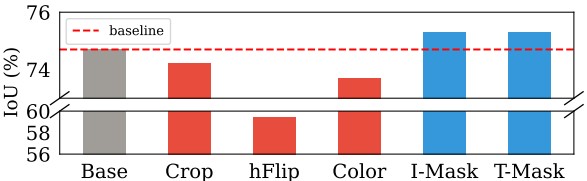

(a) Performance w/ conventional augmentations  (b) Performance w/ individual augmentation on CARIS

Figure 3: Existing RIS methods show a noticeable decline in their performance when applying conventional image augmentations (random cropping, color jittering, and horizontal flipping). In contrast, image masking (I-Mask) and text masking (T-Mask) improve model performance.

## 3  Method

In this section, we propose a novel training strategy for RIS, called Masked Referring Image Segmentation (MaskRIS). MaskRIS aims to address the limitation of previous RIS (Sec. 3.1) with two key components: (1) input masking strategy (Sec. 3.2) for data augmentation in RIS and (2) a dual-path training (Sec. 3.3) to maximize the benefits of the masking strategy.

### 3.1  Referring Image Segmentation (RIS)

**RIS Pipeline.** Given an image $x \in \mathbb{R}^{H \times W \times 3}$ and a text description specifying an object within the image, the RIS model outputs a pixel-level mask delineating the referred object. The text description is first tokenized into a sequence of words $w = \{w_1, w_2, \ldots, w_L\}$, where $w_i \in \mathbb{R}^D$ represents a word embedding with $D$ as the embedding dimension and $L$ as the number of words in the description. Here, $H$ and $W$ denote the height and width of the image, and 3 represents the RGB channels. We formalize this process as follows:

$$\hat{y} = f_\theta(x, w), \tag{1}$$

where $\hat{y} \in \mathbb{R}^{H \times W}$ is the predicted mask, generated through the interaction between the input image $x$ and the tokenized words $w$ by the neural network $f$ parameterized by $\theta$.

In conventional RIS approaches, $f$ consists of the image encoder (Liu et al., 2021b; Radford et al., 2021) and the text encoder (Devlin et al., 2018; Radford et al., 2021) to extract high-dimensional visual and language features, along with a vision-language fusion module that integrates these features to generate the segmentation mask. Training the RIS models is essentially a pixel-wise binary classification task using the cross-entropy loss function:

$$\mathcal{L}_{ce}(y, \hat{y}) = -\frac{1}{N} \sum_{i=1}^{N} y_i \log(\hat{y}_i), \tag{2}$$

where $N$ denotes the total number of pixels, $y_i$ is the binary label at each pixel $i$ in the ground truth mask (1 for object, 0 for background), and $\hat{y}_i$ is the predicted probability of the pixel belonging to the object.

**Limited Data Augmentation in RIS.** Previous studies (Yang et al., 2022; Wang et al., 2022; Liu et al., 2023c) have only focused on aligning vision and language features to bridge the modality gap. Still, effective training techniques for RIS remain underexplored, limiting the potential of vision-language features. As vision-language models grow in size and complexity (Radford et al., 2021; Jia et al., 2021; Alayrac et al., 2022), optimizing these training techniques becomes more crucial. Among training techniques, we regard data augmentation as the most promising approach, with the potential to enhance model robustness and generalization (Rebuffi et al., 2021; Li & Spratling, 2023; Yun et al., 2019; Cubuk et al., 2018; 2020).

Current RIS methods (Wang et al., 2022; Xu et al., 2023; Yang et al., 2022; Liu et al., 2023c) employ only simple resizing operations for data augmentations and don't utilize complex augmentations used in semantic segmentation. This is due to the fact that the complex data augmentations in segmentation are incompatible with RIS as shown in Figure 1, often altering or eliminating the referred objects. Consequently, these

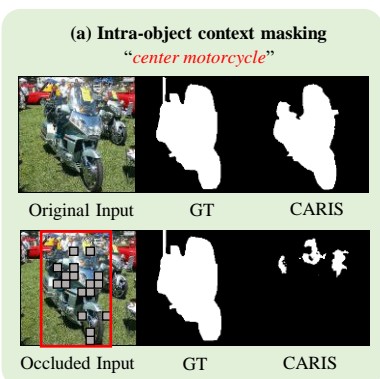 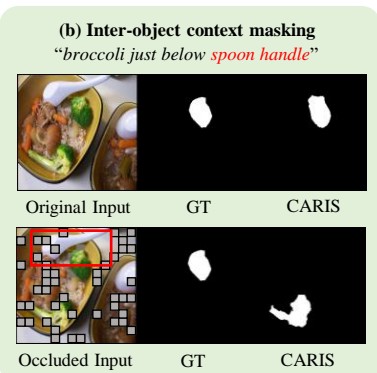 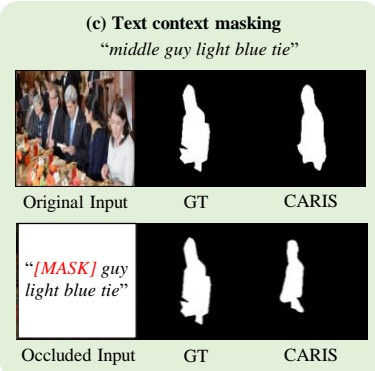

Figure 4: The existing RIS method tends to be inaccurate when faced with occluded context. CARIS (Liu et al., 2023c) represents the SoTA method in RIS. Words highlighted in red represent occluded objects in the image (left and center) and masked words in the text query (right).

augmentations lead to a decline in RIS model performance on RefCOCO as demonstrated in Figure 3. Thus, we believe tackling the limited data augmentation problem is essential for overcoming a major performance bottleneck in RIS.

## 3.2 Masking Strategy for Image and Text

To overcome the limitations, we introduce a masking-based augmentation that combines both image and text masking to generate diverse image-text training pairs. Unlike conventional data augmentation, which often distorts essential spatial and attribute details, image masking minimizes such distortions, preserving this critical information while significantly expanding data diversity. In addition, image masking is beneficial in recognizing partially visible or occluded objects, as illustrated in Figure 4(a) and (b). State-of-the-art models, such as CARIS (Liu et al., 2023c), often struggle to identify partially visible objects. This issue highlights a prevalent challenge: the presence of occlusions often leads to the failure of models trained on clear and unoccluded objects. Similarly, as seen from Figure 4(b), text descriptions often rely on contextual objects (*e.g.*, "*spoon handle*" to describe the position of "*broccoli*"), suggesting that the impact of occlusion extends beyond the target objects themselves.

Beyond managing image occlusions, RIS requires flexibility in interpreting varied text descriptions, as a single referred object can be described in multiple ways. This diversity in textual description is essential for generalizing to different referring expressions. While our image masking addresses visual occlusions, we introduce text masking to enhance the model's capability to interpret diverse and incomplete descriptions by training it to rely on broader contextual cues. By masking parts of the text, we encourage the model to infer missing details, improving its understanding and reducing dependence on specific terms. For instance, as shown in Figure 4(c), CARIS can accurately identify a "*middle guy light blue tie*" with complete text. However, it struggles with partial descriptions, such as "[MASK] *guy light blue tie*", indicating the limitations in handling incomplete textual clues. By incorporating text masking, we enable the model to adapt to various referring expressions, enhancing its comprehension and robustness in RIS tasks.

We next describe our image and text masking techniques in detail, demonstrating how they address the aforementioned challenges in RIS. In the following, we specify the detailed process for both image and text masking, clarifying how these techniques are implemented to overcome the identified challenges and enhance the RIS model's performance.

**Image Masking.** We divide the input image $x$ into non-overlapping patches and randomly mask a fixed ratio of patches. Following the MAE sampling strategy (He et al., 2022), we create a binary mask $M \in \mathbb{R}^{H \times W}$ by sampling patches uniformly without replacement, where 1 indicates a masked patch. The masked image $x^M$ is obtained by element-wise multiplication: $x^M = (1 - M) \odot x$, where $\odot$ denotes the element-wise multiplication operation. This mechanism ensures that only a subset of the image's patches is exposed to

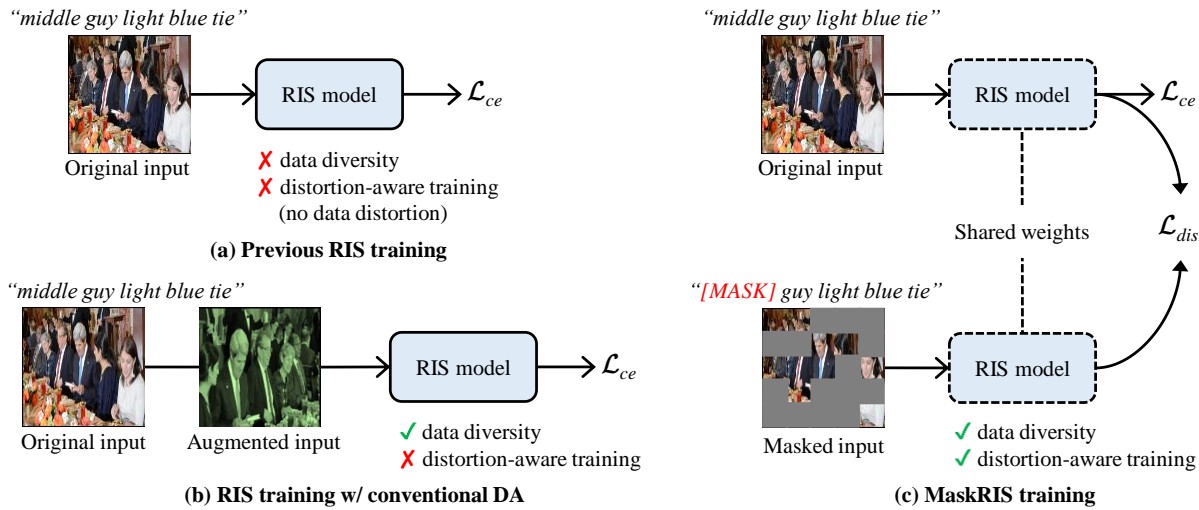

Figure 5: The overall framework of MaskRIS. Both image and text masking are employed to generate diverse image-text training pairs (Sec. 3.2). To maximize the benefits of the masking strategy, Distortion-aware Contextual Learning (DCL) is introduced (Sec. 3.3).

the model during training, compelling the neural network to infer missing information and thereby enhancing its robustness and predictive accuracy.

While there might be concerns that our approach could hinder training by completely masking target objects, such cases are extremely rare in practice. For example, even with a relatively high masking ratio of 75%, the probability of a target object being completely masked is only 0.19% - remarkably low. This probability becomes even lower with reduced masking ratios. Furthermore, this potential issue is mitigated during training as the same samples are seen multiple times. These statistics support that our masking approach effectively augments the training data while maintaining crucial visual information.

**Text Masking.** We mask word tokens to generate masked word tokens $w^M = \{w_1^M, w_2^M, \ldots, w_L^M\}$ using a probabilistic strategy in MLM (Devlin et al., 2018). Specifically, each word $w_i$ has a 15% chance of being masked. Of these, 80% are replaced with a [MASK] token, making it invisible to the model during training; 10% are replaced with a random word from the vocabulary, introducing noise and variability; and 10% remain unchanged, preserving the original context. This strategy allows the model to experience a variety of textual scenarios, enhancing its ability to comprehend context and meaning even when parts of the text are masked or altered. Notably, as Wei & Zou (2019) witnessed, random word masking effectively preserves sentence meaning, further validating our method's balance between sentence coherence and data diversity.

### 3.3 Distortion-aware Contextual Learning

In the training phase, we employ both image and text masking simultaneously, enriching the diversity of the training dataset. Specifically, we formalize the processing of masked inputs through the model as follows:

$$\hat{y}^M = f_\theta(x^M, w^M), \tag{3}$$

where $\hat{y}^M$ denotes the model's output for the masked inputs. By adopting this strategy, MaskRIS learns to effectively segment objects under varying conditions of occlusion and incomplete descriptions. This simple integration effectively mitigates the intrinsic challenge of RIS that stems from the absence of an effective data augmentation strategy.

To further facilitate the RIS training, we introduce Distortion-aware Contextual Learning (DCL), as shown in Figure 5. This approach uses a primary path for processing original inputs and a secondary path for masked inputs. The primary path focuses on preserving the original image context, ensuring training stability, and promoting distortion-aware training. Meanwhile, the secondary path introduces variability by processing

masked data, which improves the model's robustness. The training objective of MaskRIS consists of two parts, the distillation loss $\mathcal{L}_{dist}$ and the original pixel-wise classification loss $\mathcal{L}_{ce}$ in Eq. 2, *i.e.*, $\mathcal{L}_{total} = \mathcal{L}_{dist} + \mathcal{L}_{ce}$. In practice, we adopt a weighting factor between $\mathcal{L}_{ce}$ and $\mathcal{L}_{dist}$. While our default setting uses $\lambda = 0.5$, we also provide a detailed sensitivity analysis of this ratio in Appendix A.5, showing that MaskRIS is robust across a wide range of choices.

The distillation loss is defined through the binary cross-entropy loss between the predictions from both original and masked inputs as follows:

$$\mathcal{L}_{dist} = \mathcal{L}_{ce}(\mathsf{sg}(\hat{y}), \hat{y}^M) = -\frac{1}{N} \sum_{i=1}^{N} \hat{y}_i \log(\hat{y}_i^M), \tag{4}$$

where $\mathsf{sg}$ is the stop-gradient function, which blocks gradient back-propagation. Inspired by Heo et al. (2023), MaskRIS employs a self-distillation framework that leverages soft targets from the original inputs, reducing training complexity and enhancing model generalization. This approach allows MaskRIS to effectively mitigate the learning instability caused by input masking while maintaining the benefits of increased data diversity and robustness.

## 4 Experiments

### 4.1 Experimental Setups

**Datasets.** We evaluate our method using three popular benchmarks in RIS: RefCOCO, RefCOCO+, and RefCOCOg. RefCOCO (Yu et al., 2016), all built on the MS COCO dataset (Lin et al., 2014). RefCOCO provides 142,209 expressions for about 50,000 objects across 19,994 images, emphasizing object locations and appearances. RefCOCO+ (Yu et al., 2016) offers over 141,564 expressions, excluding spatial language to focus on object attributes. RefCOCOg (Yu et al., 2016) introduces more complexity into the evaluation by including 85,474 longer expressions (8.43 words per expression on average) for 54,822 objects in 26,711 images; for this dataset, we report results on the UMD partition (Yu et al., 2016), following the previous studies (Wang et al., 2022; Liu et al., 2023b; Kim et al., 2023b). To further examine cross-dataset generalization beyond COCO-style images, we additionally evaluate MaskRIS on RefClef, a subset of the ImageCLEF dataset with more diverse natural scenes and object categories (see Appendix A.1 for details).

**Evaluation Metrics.** To evaluate model performance, we use mean Intersection-over-Union (mIoU), overall Intersection-over-Union (oIoU), and P@X. While mIoU calculates the average IoU across all test samples, oIoU measures the ratio of the total intersection to the total union areas across all samples. P@X measures the percentage of test samples with an IoU above a threshold $\mathsf{X} \in \{0.5, 0.7, 0.9\}$. These metrics ensure a consistent and fair evaluation across different RIS methods.

**Implementation Details.** To validate MaskRIS, we applied it to various RIS models (Xu et al., 2023; Wang et al., 2022; Yang et al., 2022; Liu et al., 2023c). Designed as a plug-and-play training strategy, we strictly follow the original training settings and hyperparameters, such as learning rate, epochs, and batch size, without modification. Notably, we primarily implemented our method on CARIS (Liu et al., 2023c), a leading SoTA method, unless stated otherwise. Images are resized to $448 \times 448$ for both training and testing. For image masking, we set 32 as the patch size.

### 4.2 Main Results

**Comparison with State of the Arts.** We compare MaskRIS with previous methods on three popular benchmarks. As shown in Table 1, MaskRIS consistently outperforms previous methods by significant margins. Specifically, compared to the second-best performing method, CARIS (Liu et al., 2023c), our approach improves oIoU scores by 1.82%p, 1.33%p, and 2.25%p on the RefCOCO validation, testA, and testB sets, respectively. On RefCOCO+, our method leads by 1.37%p, 2.76%p, and 1.93%p on the validation, testA, and testB splits, respectively. Even on the challenging RefCOCOg dataset, MaskRIS still outperforms CARIS by 0.89%p and 1.05%p on the validation and test splits. These results validate the effectiveness of our masking strategy for RIS model training.

| Method | Image Encoder | Text Encoder | RefCOCO | | | RefCOCO+ | | | RefCOCOg | |
|---|---|---|---|---|---|---|---|---|---|---|
| | | | val | testA | testB | val | testA | testB | val | test |
| *Standard: Training on the training split of each dataset.* | | | | | | | | | | |
| **mIoU** | | | | | | | | | | |
| CRIS (Wang et al., 2022) | RN101 | CLIP | 70.47 | 73.18 | 66.10 | 62.27 | 68.08 | 53.68 | 59.87 | 60.36 |
| ETRIS (Xu et al., 2023) | RN101 | CLIP | 71.06 | 74.11 | 66.66 | 62.23 | 68.51 | 52.79 | 60.28 | 60.42 |
| RefTR (Li & Sigal, 2021) | RN101 | BERT | 74.34 | 76.77 | 70.87 | 66.75 | 70.58 | 59.40 | 66.63 | 67.39 |
| LAVT (Yang et al., 2022) | Swin-B | BERT | 74.46 | 76.89 | 70.94 | 65.81 | 70.97 | 59.23 | 63.34 | 63.62 |
| CM-MaskSD (Wang et al., 2024) | ViT-L | CLIP | 74.89 | 77.54 | 71.28 | 67.47 | 71.80 | 59.91 | 66.53 | 66.63 |
| CGFormer (Tang et al., 2023) | Swin-B | BERT | 76.93 | 78.70 | 73.32 | 68.56 | 73.76 | 61.72 | 67.57 | 67.83 |
| MaskRIS | Swin-B | BERT | **78.35** | **80.24** | **76.06** | **71.68** | **76.73** | **64.50** | **69.31** | **69.42** |
| **oIoU** | | | | | | | | | | |
| LAVT (Yang et al., 2022) | Swin-B | BERT | 72.73 | 75.82 | 68.79 | 62.14 | 68.38 | 55.10 | 61.24 | 62.09 |
| CGFormer (Tang et al., 2023) | Swin-B | BERT | 74.75 | 77.30 | 70.64 | 64.54 | 71.00 | 57.14 | 64.68 | 65.09 |
| LQMFormer (Shah et al., 2024) | Swin-B | BERT | 74.16 | 76.82 | 71.04 | 65.91 | 71.84 | 57.59 | 64.73 | 66.04 |
| NeMo (Ha et al., 2024) | Swin-B | BERT | 74.48 | 76.32 | 71.51 | 62.86 | 69.92 | 55.56 | 64.40 | 64.80 |
| ReMamber (Yang et al., 2024) | Mamba-B | CLIP | 74.54 | 76.74 | 70.89 | 65.00 | 70.78 | 57.53 | 63.90 | 64.00 |
| CARIS† (Liu et al., 2023c) | Swin-B | BERT | 74.67 | 77.63 | 71.71 | 66.17 | 71.70 | 57.46 | 64.66 | 65.45 |
| MagNet (Chng et al., 2024) | Swin-B | BERT | 75.24 | 78.24 | 71.05 | 66.16 | 71.32 | 58.14 | 65.36 | 66.03 |
| MaskRIS | Swin-B | BERT | **76.49** | **78.96** | **73.96** | **67.54** | **74.46** | **59.39** | **65.55** | **66.50** |
| *Combined: Training on the combination of three datasets.* | | | | | | | | | | |
| **oIoU** | | | | | | | | | | |
| PolyFormer (Liu et al., 2023b) | Swin-B | BERT | 74.82 | 76.64 | 71.06 | 67.64 | 72.89 | 59.33 | 67.76 | 69.05 |
| CARIS† (Liu et al., 2023c) | Swin-B | BERT | 76.67 | 79.85 | 73.07 | 68.23 | 73.98 | 59.59 | 67.42 | 68.73 |
| MaskRIS | Swin-B | BERT | **78.71** | **80.64** | **75.10** | **70.26** | **75.15** | **62.83** | **69.12** | **71.09** |

Table 1: Comparison with state-of-the-art methods on three benchmark datasets. † denotes the reproduced results across all experiments.

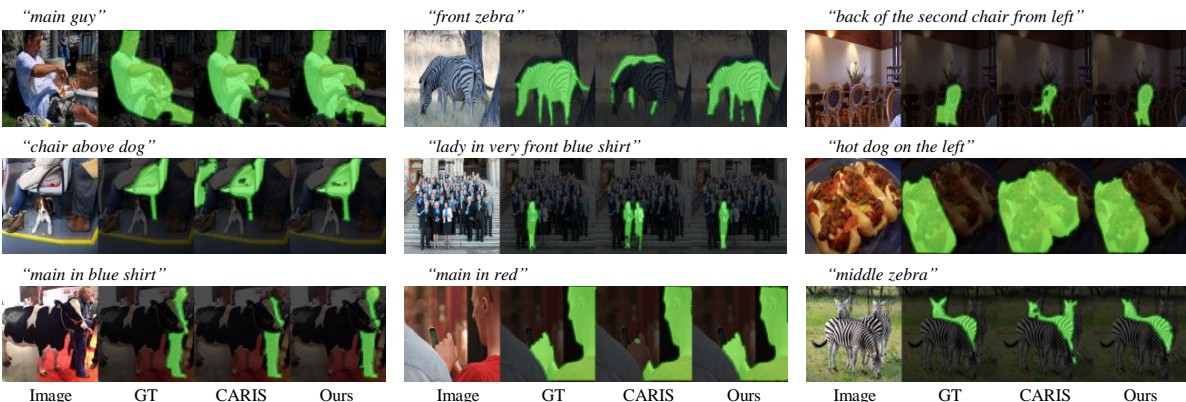

Figure 6: Qualitative examples of segmentation results on the RefCOCO dataset.

To further demonstrate the capability of our method, we conduct experiments on a larger, more comprehensive dataset. This dataset is a combination of the training sets from RefCOCO, RefCOCO+, and RefCOCOg, following the approach used in previous studies (Liu et al., 2023b;c). Training on this combined dataset yields even better results, with oIoU improvements of 1.17–4.04%p over previous methods. This result indicates that our method is effective across different types of data and highlights its generalizability. Figure 6 shows qualitative examples on RefCOCO, where our method more comprehensively captures the extent of objects by expanding or reducing their coverage (1st row and 2nd row). Additionally, it demonstrates robustness against occlusion (3rd row) and precisely identifies target objects in alignment with the provided text descriptions (4th row). These results confirm the ability of our approach to enhance model robustness and leverage textual cues effectively.

**Robustness Evaluation.** A key strength of our approach lies in its robustness across visually and linguistically complex scenarios. To this end, we first evaluate our model's robustness to various image corruptions. We use the benchmark provided by ImageNet-C (Hendrycks & Dietterich, 2019), which includes 17 types of corruption, each with 5 severity levels. To evaluate the performance, we compute the oIoU at each severity level and average these scores for each corruption type. As shown in Figure 7(a), MaskRIS consistently out-

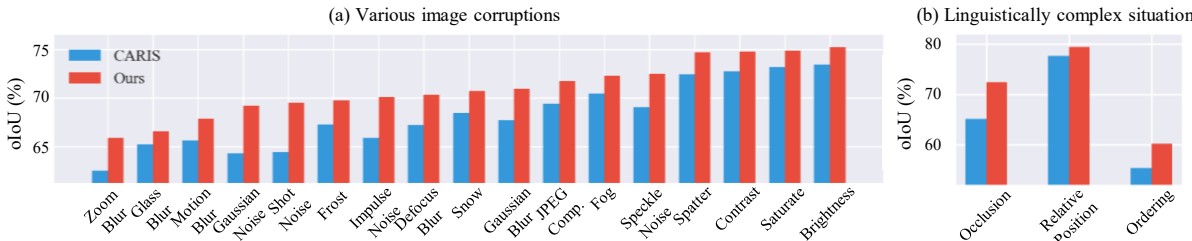

Figure 7: Robustness on various image corruptions provided by ImageNet-C (Hendrycks & Dietterich, 2019) and linguistically complex situations. The results (oIoU) are evaluated on the RefCOCO validation set.

| Method | Image Encoder | Text Encoder | RefCOCO | | |
|---|---|---|---|---|---|
| | | | val | testA | testB |
| *Weakly-supervised approach* | | | | | |
| **mIoU** | | | | | |
| TRIS (Liu et al., 2023a) | RN50 | CLIP | 31.17 | 32.43 | 29.56 |
| TRIS (Liu et al., 2023a)+MaskRIS | RN50 | CLIP | 32.60 (+1.43) | 34.26 (+1.83) | 30.32 (+0.76) |
| SaG (Kim et al., 2023a) | ViT-S/16 | BERT | 37.21 | 36.60 | 39.41 |
| SaG (Kim et al., 2023a)+MaskRIS | ViT-S/16 | BERT | 38.85 (+1.64) | 37.70 (+1.1) | 40.20 (+0.79) |
| *Fully-supervised approach* | | | | | |
| **mIoU** | | | | | |
| CRIS (Wang et al., 2022) | RN50 | CLIP | 69.52 | 72.72 | 64.70 |
| CRIS (Wang et al., 2022)+MaskRIS | RN50 | CLIP | 70.73 (+1.21) | 74.06 (+1.34) | 66.82 (+2.12) |
| ETRIS (Xu et al., 2023) | R101 | CLIP | 71.06 | 74.11 | 66.66 |
| ETRIS (Xu et al., 2023)+MaskRIS | R101 | CLIP | 72.39 (+1.33) | 74.99 (+0.88) | 68.55 (+1.89) |
| **oIoU** | | | | | |
| LAVT (Yang et al., 2022) | Swin-B | BERT | 72.73 | 75.82 | 68.79 |
| LAVT (Yang et al., 2022)+MaskRIS | Swin-B | BERT | 73.79 (+1.06) | 76.57 (+0.75) | 70.31 (+1.52) |
| ReMamber[†] (Yang et al., 2024) | Mamba-B | CLIP | 72.37 | 74.85 | 68.29 |
| ReMamber (Yang et al., 2024)+MaskRIS | Mamba-B | CLIP | 74.02 (+1.65) | 76.32 (+1.47) | 69.06 (+0.77) |
| DETRIS[†] (Huang et al., 2025) | DINOv2-B | CLIP | 74.47 | 77.40 | 71.46 |
| DETRIS (Huang et al., 2025)+MaskRIS | DINOv2-B | CLIP | 76.08 (+1.61) | 78.46 (+1.06) | 72.84 (+1.38) |
| CARIS[†] (Liu et al., 2023c) | Swin-B | BERT | 74.67 | 77.63 | 71.71 |
| CARIS (Liu et al., 2023c)+MaskRIS | Swin-B | BERT | 76.49 (+1.82) | 78.96 (+1.33) | 73.96 (+2.25) |

Table 2: Compatibility of MaskRIS with various RIS methods. MaskRIS consistently enhances existing methods in both fully supervised and weakly supervised settings.

performs the previous SoTA method, CARIS (Liu et al., 2023c), with oIoU improvements of 1.34–5.07%p. This demonstrates that MaskRIS effectively reduces overfitting to clean data, providing greater robustness against a wide range of visual distortions.

In addition, we evaluate MaskRIS's robustness to linguistically complex situations, such as occlusion, relative position, and ordering. For occlusion scenarios, we modify the input data by occluding parts of objects. For relative position and ordering, we select samples containing relevant linguistic keywords from the validation set. As shown in Figure 7(b), MaskRIS achieves superior performance in these complex scenarios on the RefCOCO validation set, further highlighting its robustness to both visual and linguistic complexity.

**Compatibility with Other Methods.** Our training strategy is highly compatible, integrates seamlessly with various RIS methods, and achieves significant performance improvements across both weakly supervised and fully supervised settings, as shown in Table 2. In the weakly supervised regime, where models rely solely on textual descriptions without ground-truth masks, MaskRIS still yields notable gains. For instance, TRIS (Liu et al., 2023a) improves by +1.43%p on the validation set, +1.83%p on testA, and +0.76%p on testB. Likewise, SaG (Kim et al., 2023a) is enhanced by up to +1.64%p, reaching 38.85 mIoU on the validation set. In the fully supervised setting, MaskRIS demonstrates equally strong compatibility. CRIS (Wang et al., 2022) shows improvements of +1.21%p (val), +1.34%p (testA), and +2.12%p (testB), while ETRIS (Xu et al., 2023) gains up to +1.89%p. Beyond CNN-based baselines, transformer-based approaches such as

| Method | Image Encoder | Text Encoder | RefCOCO | | |
|---|---|---|---|---|---|
| | | | val | testA | testB |
| **DetAcc** | | | | | |
| EEVG[†] (Chen et al., 2024) | ViT-B | BERT | 87.87 | 89.97 | 85.34 |
| EEVG (Chen et al., 2024)+MaskRIS | ViT-B | BERT | 88.72 (+0.85) | 90.81 (+0.84) | 85.36 (+0.02) |
| SimVG[†] (Dai et al., 2024) | ViT-B | BEiT-3 | 86.94 | 88.97 | 82.87 |
| SimVG (Dai et al., 2024)+MaskRIS | ViT-B | BEiT-3 | 87.79 (+0.85) | 90.02 (+1.05) | 84.88 (+2.01) |
| C3VG[†] (Dai et al., 2025) | ViT-B | BEiT-3 | 92.42 | 94.12 | 89.34 |
| C3VG (Dai et al., 2025)+MaskRIS | ViT-B | BEiT-3 | 93.29 (+0.87) | 94.86 (+0.74) | 90.29 (+0.95) |

Table 3: Compatibility of MaskRIS with REC methods. MaskRIS consistently enhances existing methods.

| IM | TM | DCL | P@0.5 | P@0.7 | P@0.9 | oIoU |
|---|---|---|---|---|---|---|
| ✘ | ✘ | ✘ | 87.73 | 80.20 | 39.60 | 74.67 |
| ✔ | ✘ | ✘ | 87.76 | 80.52 | 38.73 | 75.31 |
| ✘ | ✔ | ✘ | 87.72 | 80.45 | 39.26 | 75.32 |
| ✔ | ✔ | ✘ | 88.00 | 81.35 | 40.11 | 75.71 |
| ✔ | ✘ | ✔ | 88.60 | 81.86 | 41.08 | 76.02 |
| ✘ | ✔ | ✔ | 88.34 | 81.19 | 39.24 | 75.76 |
| ✔ | ✔ | ✔ | **88.62** | **81.95** | **41.19** | **76.49** |

Table 4: Impact of each component of MaskRIS on the RefCOCO validation set, where IM (TM) refers to image (text) masking.

LAVT (Yang et al., 2022) and CARIS (Liu et al., 2023c) also benefit significantly, with CARIS achieving the highest improvements of +2.25%p on testB. More recent architectures, including ReMamber (Yang et al., 2024) and DETRIS (Huang et al., 2025), further validate the generality of our method, consistently obtaining +0.8–2.0%p gains without architectural modifications. Overall, MaskRIS consistently improves RIS model performance across diverse architectures and training paradigms, demonstrating its adaptability and effectiveness without modifying the original designs.

**Compatibility with REC Methods.** The conflict between conventional image augmentations and language-grounded tasks is not unique to RIS. Referring Expression Comprehension (REC) also relies on precise spatial and attribute alignment between text and image. For example, flipping an image containing the phrase "the man on the left" alters the semantic meaning for both RIS and REC. This confirms that the augmentation dilemma is indeed universal to referring tasks. To investigate this, we integrated MaskRIS into various REC methods (Chen et al., 2024; Dai et al., 2024; 2025). As shown in Table 3, MaskRIS consistently improves performance when combined with existing REC models by enhancing robustness to occlusion and incomplete linguistic cues while increasing data diversity. The findings confirm that MaskRIS is not task-specific. Its masking approach addresses the augmentation dilemma across both RIS and REC, making it a general training strategy for referring tasks where traditional augmentations are unsafe.

### 4.3 Ablation Study

**Impact of Each Component of MaskRIS.** We analyze the impact of each MaskRIS component in model training. Table 4 shows segmentation results on the RefCOCO validation set. The results clearly show that masking images and text separately improves model performance. However, the most significant improvement occurs when we apply masking to both images and text together. This demonstrates that the combination of visual and textual masking strategies significantly improves the model's ability to accurately identify and segment the referred objects. In addition, Distortion-aware Contextual Learning (DCL) provides further performance gains. It effectively mitigates the learning instability caused by input masking, while maintaining the benefits of increased data diversity and robustness.

Figure 8 illustrates the benefits of image and text masking. CARIS often fails to correctly identify objects when they or their adjacent objects are partially obscured. However, by incorporating image masking, our approach achieves accurate object recognition (1st and 2nd rows). For example, in the scenario shown

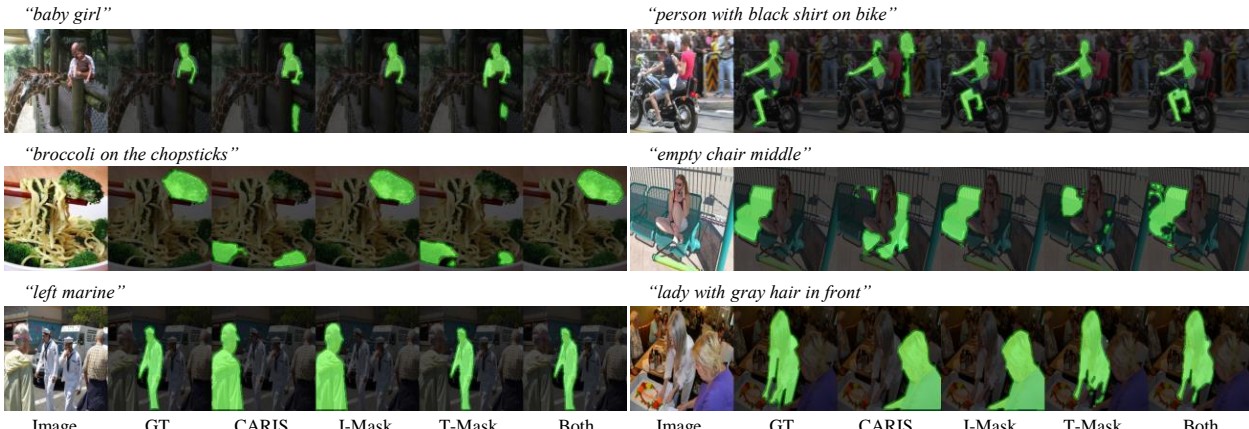

Figure 8: Qualitative examples of masking strategies on the RefCOCO dataset. I-Mask, T-Mask, and Both denote the results of applying image masking, text masking, and both, respectively.

in the first row with a *"baby girl"* whose legs are obscured, CARIS inaccurately identifies the legs of an adult as part of the target object. On the other hand, our method successfully separates the obscured elements, demonstrating the superior accuracy of our method in complex environments. Conversely, text masking improves the alignment between the text description and the target object (3rd row). For example, with the *"left marine"*, CARIS inaccurately identifies the target object by focusing too much on spatial information (*i.e.*, *"left"*), as shown in Figure 4(c). Text masking intervenes in such cases, ensuring the accurate recognition of the target object. Furthermore, by integrating both image and text masking, our approach exploits the strengths of each masking strategy.

### 4.4 Discussion: Comparison with CM-MaskSD

MaskRIS differs from CM-MaskSD (Wang et al., 2024) in three key aspects: purpose, masking strategy, and adaptability. First, MaskRIS explicitly handles the semantic conflicts caused by conventional augmentations in RIS, ensuring that transformations do not alter the contextual meaning of referring expressions. In contrast, CM-MaskSD focuses on improving patch-text alignment using CLIP's global vector but does not directly tackle

| Masking | Image Enc. | Text Enc. | oIoU |
|---|---|---|---|
| Object-centric | | | 75.14 |
| Context | Swin-B | BERT | 75.93 |
| Holistic | | | **76.02** |

Table 5: Comparison of image masking approaches on the RefCOCO validation set.

augmentation-induced inconsistencies. This makes semantic conflict resolution a unique contribution of MaskRIS. Second, while CM-MaskSD employs object-centric masking, selectively targeting key objects or keywords, MaskRIS utilizes a holistic masking approach that applies masking across all image regions. This strategy enhances model generalization and robustness. As shown in Table 5, holistic masking consistently outperforms object-centric masking. Third, CM-MaskSD is architecture-dependent due to its reliance on the [CLS] token, limiting its applicability to CLIP-based models. In contrast, MaskRIS is architecture-agnostic and can be seamlessly integrated into various RIS and REC frameworks.

## 5 Conclusion

In this study, we introduce Masked Referring Image Segmentation (MaskRIS), an effective training framework for Referring Image Segmentation (RIS) that combines image and text masking with Distortion-aware Contextual Learning (DCL). MaskRIS addresses the challenges of conventional data augmentation in RIS by minimizing semantic distortion and enhancing data diversity. This approach strengthens the model's robustness to occlusions and incomplete information, achieving new state-of-the-art accuracies on the RefCOCO, RefCOCO+, and RefCOCOg datasets. Our results demonstrate MaskRIS's effectiveness and adaptability in both fully and weakly-supervised settings, highlighting its potential as a versatile and powerful baseline for advancing the field of RIS.

## Acknowledgments

Some parts of experiments are based on the NAVER Smart Machine Learning NSML (Kim et al., 2018) platform. This research was supported by Institute of Information & communications Technology Planning & Evaluation (IITP) grant funded by the Korea government (MSIT) (RS-2025-02217259, 25%), the Basic Science Research Program through the National Research Foundation of Korea (NRF) funded by the MSIP (RS-2025-00520207, 25%, RS-2023-00219019, 25%), and Samsung Electronics Co., Ltd (IO230508-06190-01, 25%).

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

# *Supplementary Material*

## Contents

## A  Additional Experiments

| Method | Image Encoder | Text Encoder | oIoU |
|---|---|---|---|
| CARIS (Liu et al., 2023c) | Swin-B | BERT | 75.21 |
| CARIS + MaskRIS | | | **76.06** |

Table A: Comparison on RefClef (ImageCLEF) dataset. MaskRIS consistently improves over the baseline, demonstrating generalization beyond MS COCO benchmarks.

### A.1  Evaluation on RefClef

Following the common practice of prior works in RIS, our main experiments are conducted on MS COCO–based benchmarks (RefCOCO, RefCOCO+, RefCOCOg). In addition, we validated MaskRIS on RefClef (Kazemzadeh et al., 2014), which is a subset of the ImageCLEF dataset. Unlike MS COCO, RefClef includes more diverse natural scenes and a broader set of object categories beyond MS-COCO's predefined classes.

As shown in Table A, MaskRIS outperforms the CARIS baseline (76.06 vs. 75.21), confirming that its effectiveness is not limited to COCO-style data. These results demonstrate that our holistic context masking and distortion-aware training framework generalizes well across datasets with varying scene compositions and category coverage, thereby reinforcing the broader applicability of MaskRIS.

### A.2  Analysis of Image Masking Strategy

To rigorously validate the impact of image masking on model training, we explored various mask sampling strategies and their effects. Figure A visually illustrates each method. Our baseline, patch-wise sampling (Heo et al., 2023; Hoyer et al., 2023; He et al., 2022) divides images into non-overlapping patches and randomly masks certain patches. This approach is widely used due to its simplicity and effectiveness. We

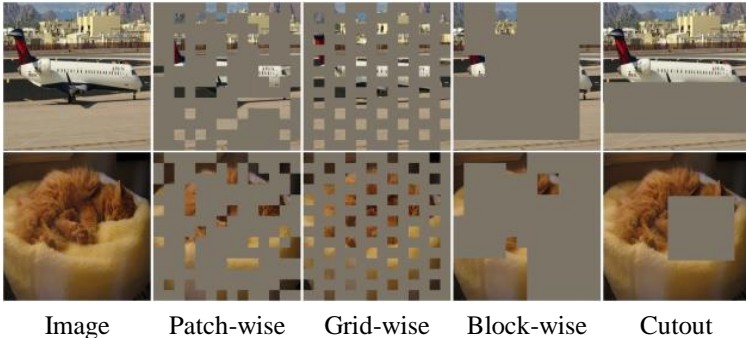

| | Image | Patch-wise | Grid-wise | Block-wise | Cutout |

Figure A: Various types of image mask strategies. We use patch-wise sampling (He et al., 2022) as our default. For block-wise sampling, we follow BEiT(Bao et al., 2021) to remove large random blocks. For Cutout (DeVries & Taylor, 2017), we follow the implementation of timm (Wightman, 2019). For all strategies except Cutout, we maintain a consistent masking ratio of 75%.

| | Patch-wise | Grid-wise | Block-wise | Cutout |
|---|---|---|---|---|
| oIoU | **76.49** | 76.14 | 75.90 | 76.07 |

Table B: Impact of imaeg masking strategy. The performance (oIoU) is evaluated across various masking strategies on the RefCOCO validation set.

| | MaskRIS | WWM | SpanBERT | POS-based |
|---|---|---|---|---|
| oIoU | 76.49 | 76.03 | 76.46 | **76.52** |

Table C: Impact of text masking strategy. The performance (oIoU) is evaluated across various masking strategies on the RefCOCO validation set.

also tested grid-wise sampling (He et al., 2022), where rows and columns are masked systematically rather than randomly. Additionally, we investigated block-wise masking, as used in BEiT (Bao et al., 2021), where contiguous regions of the image are masked, potentially eliminating more important structural details by covering larger areas. Finally, we evaluated Cutout (DeVries & Taylor, 2017), which masks a randomly selected rectangular section of the image.

Table B summarizes the results (oIoU) of each mask sampling strategy on the RefCOCO validation set. Remarkably, all the strategies we tested consistently outperform the previous SoTA method, CARIS (Liu et al., 2023c) (74.67 oIoU), with significant margins. Among these strategies, our patch-wise random sampling approach achieved superior performance, demonstrating its ability to generate more varied training data than grid-wise sampling and minimize training data distortion more effectively than block-wise sampling. These results confirm the importance of selecting an appropriate masking strategy to improve performance.

### A.3 Analysis of Text Masking Strategy

To further validate the role of linguistic perturbation, we conducted an ablation study on various text masking strategies. Specifically, we compared four schemes. (1) Token-level random masking, which is adopted in MaskRIS. (2) Whole-Word Masking (WWM) (Cui et al., 2021), which masks all sub-tokens of a sampled word (*e.g.*, masking "playing" removes both "play" and "##ing"), often hiding entire entities or key terms. (3) SpanBERT-style masking (Joshi et al., 2020), which replaces contiguous spans of tokens (*e.g.*, masking a two-word phrase like "on the"). This encourages the model to capture longer-range contextual dependencies. (4) Part-of-Speech (POS)-aware random masking, where the target POS is treated as a tunable parameter. In our setup, entity-bearing nouns and relational terms are preserved, while adjectives and adverbs are randomly masked. This approach recognizes the importance of entities while introducing variation through modifiers.

| Method | val | testA | testB |
|---|---|---|---|
| DCL (Ours) | 76.49 | **78.96** | **73.96** |
| Cross-distillation | **76.68** | 78.92 | 73.21 |

Table D: Impact of distillation strategy. The performance (oIoU) is evaluated across various distillation strategies on RefCOCO.

| ps | 8 | 16 | 32 | 64 | 112 |
|---|---|---|---|---|---|
| oIoU | **76.58** | 76.49 | 76.49 | 76.47 | 76.01 |

Table E: Impact of patch size on image masking. The performance (oIoU) is evaluated across different patch sizes on the RefCOCO validation set. `ps` denotes the patch size.

| $\lambda$ | 0.1 | 0.25 | 0.5 | 0.75 | 1.0 |
|---|---|---|---|---|---|
| oIoU | 75.48 | **76.62** | 76.49 | 76.16 | 74.67 |

Table F: Impact of $\lambda$ on DCL procedure. The performance (oIoU) is evaluated across various $\lambda$ on the RefCOCO validation set.

Table C summarizes the performance on the RefCOCO validation set. All four strategies outperform the previous SoTA method, CARIS (Liu et al., 2023c) (74.67 oIoU), with consistent margins. Among them, WWM tends to remove entire entities, leading to slightly lower performance (76.03 oIoU). Span masking performs comparably to token-level masking (76.46 vs. 76.49). The POS-aware strategy achieves the best result (76.52), highlighting the importance of preserving semantic anchors while introducing controlled lexical noise. These findings, combined with our image masking study, demonstrate that carefully selecting what to mask in both visual and textual modalities is critical to achieving robust improvements.

### A.4 Analysis of Distillation Strategy

We further investigated whether incorporating a cross-modal contrastive loss (Wang et al., 2024) could enhance our distortion-aware contextual learning (DCL) framework. As shown in Table D, adding cross-distillation yields a slight gain on the validation set (76.68 vs. 76.49), but this improvement does not generalize to the test splits. On RefCOCO testA, performance remains almost identical (78.92 vs. 78.96), and on testB it even decreases (73.21 vs. 73.96). In addition, this setting substantially increases computational cost, as each training step requires three forward passes instead of one. These results confirm that cross-distillation does not provide consistent accuracy benefits and results in an unfavorable trade-off between performance and efficiency.

### A.5 Impact of Hyperparameters

**Patch Size of Image Masking.** To further analyze the hyperparameters of image masking, we examined the training results of different patch sizes for patch-wise image masking. Table E shows that our method consistently achieves superior performance regardless of patch size, outperforming the current SoTA method, CARIS (Liu et al., 2023c). Specifically, we observe that a patch size of 8 yields the highest performance. Nevertheless, the results suggest that our method exhibits stable and strong performance as long as the patch size remains within a reasonable range, highlighting its robustness to patch size variations.

**Loss Weight of DCL.** As described in Sec. 3.3, our training objective of Distortion-aware Contextual Learning (DCL) is $\mathcal{L}_{total} = \mathcal{L}_{dist} + \mathcal{L}_{ce}$. To balance the scale of the loss values with previous methods (Liu et al., 2023c) during the training phase, we adapt the equation to:

$$\mathcal{L}_{total} = (1 - \lambda) \cdot \mathcal{L}_{dist} + \lambda \cdot \mathcal{L}_{ce}, \tag{5}$$

where $\lambda$ is set to 0.5 for all experiments to avoid extensive hyperparameter search for optimal performance. A $\lambda$ of 1.0 indicates training only with $\mathcal{L}_{ce}$, which is the equivalent to CARIS (Liu et al., 2023c). In Table F,

| IM ratio | TM ratio | $p_m$ | $p_r$ | $p_u$ | oIoU |
|---|---|---|---|---|---|
| - | 0.15 | 0.8 | 0.1 | 0.1 | 75.76 |
| - | 0.15 | 0.8 | 0.2 | 0 | 75.43 |
| - | 0.15 | 0.5 | 0.5 | 0 | 75.70 |
| - | 0.50 | 0.5 | 0.5 | 0 | 74.88 |
| - | 0.75 | 0.5 | 0.5 | 0 | 74.14 |
| 0.25 | 0.15 | 0.8 | 0.1 | 0.1 | 76.07 |
| 0.50 | 0.15 | 0.8 | 0.1 | 0.1 | 76.43 |
| 0.75 | 0.15 | 0.8 | 0.1 | 0.1 | **76.49** |

Table G: Impact of the masking ratio on the RefCOCO validation set. When a word $w_i$ is selected based on the masking ratio, it has a $p_m$ probability of being masked with a [MASK] token, a $p_r$ probability of being replaced with a random word from the vocabulary, and a $p_u$ probability of remaining unchanged.

| Method | Epoch | val | testA | testB |
|---|---|---|---|---|
| CARIS[†] (Liu et al., 2023c) | 50 | 74.67 | 77.63 | 71.71 |
| CARIS[†] (Liu et al., 2023c) | 100 | 74.80 | 77.38 | 70.63 |
| MaskRIS | 25 | 75.87 | 77.99 | 73.51 |
| MaskRIS | 50 | **76.49** | **78.96** | **73.96** |

Table H: Comparison of computational cost. All results (oIoU) are evaluated on the RefCOCO dataset. MaskRIS outperforms CARIS even with half of the training epochs.

we explore how varying $\lambda$ affects the DCL procedure. As shown in Table F, while the best performance is achieved with $\lambda = 0.25$, the performance with $\lambda$ of 0.5 remains significantly better than the results without $\mathcal{L}_{dist}$. These results confirm that MaskRIS does not require extensive hyperparameter search for $\lambda$ and performs reasonably well with $\lambda$ of 0.5.

**Impact of Masking Ratio.** We investigate how masking ratios affect model performance, focusing on the balance between masking images and text. Table G provides insights into the optimal use of masking in model training. For text masking, we follow the masking ratio used by BERT (Devlin et al., 2018). Note that we refer to the BERT setting as it is, setting $p_u$ to 10% is equivalent to combining $p_m$ to 89% with $p_r$ to 11%. We observe a significant drop in performance as the text masking ratio increases, indicating that removing too much textual information interferes with model training. On the other hand, image masking shows an optimal improvement at a masking ratio of 75%, suggesting that a higher degree of image masking improves model performance. These observations highlight the balance of masking ratios across different modalities. Excessive text masking can negatively affect learning by causing the loss of crucial information, whereas increasing image masking up to a certain threshold can be beneficial.

## A.6 Analysis of Computational Cost.

MaskRIS needs additional computational costs due to the dual-path framework, which requires forwarding both the original and masked inputs. For a fair comparison, we adjust the training epochs and examine the impact on performance. As shown in Table H, doubling the training epochs for CARIS does not enhance its performance, indicating that MaskRIS's improved performance is not a result of increased computation. Moreover, even when our method is trained for only 25 epochs, it still outperforms CARIS trained for 50 epochs. This highlights the efficacy of our masking strategy and dual-path approach in boosting performance efficiently. Notably, since MaskRIS does not change the model architecture or add additional parameters, it maintains the same inference time as the baseline.

## A.7 Analysis of Training Loss.

We validate the effectiveness of MaskRIS through an in-depth analysis of training loss, comparing three distinct settings: CARIS (Liu et al., 2023c), CARIS w/ Masking, and MaskRIS. For CARIS w/ Masking, we employ both image and text masking as data augmentation strategies, applying them with a 50% probability. Figure B shows the training loss and the model performance changes over various training epochs. In

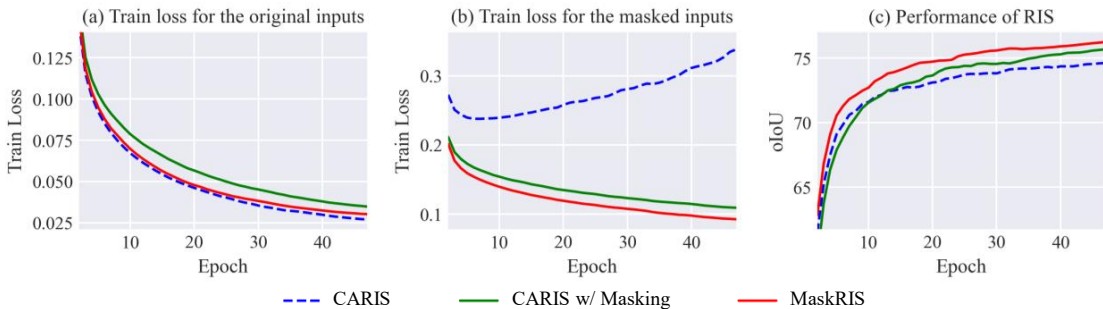

Figure B: Training loss analysis of MaskRIS on RefCOCO. In the CARIS w/ Masking setting, we employ both image and text masking as data augmentation strategies within the CARIS (Liu et al., 2023c) framework. We visualize (a) the training loss for the original (*i.e.*, unmasked) inputs, (b) the training loss for the masked inputs, and (c) the performance of the model on the RefCOCO validation set.

the baseline CARIS, training loss with the original inputs rapidly converges, while training loss with the masked inputs increases. This indicates that CARIS overfits to clean inputs as training progresses. In contrast, CARIS w/ Masking effectively addresses this overfitting by incorporating input masking. However, it degrades loss convergence and disrupts learning stability. Our MaskRIS improves loss convergence for both original and masked inputs and at the same time significantly increases model performance.

## B   More Qualitative Examples

Figure C provides additional qualitative examples on the RefCOCO dataset across different occlusion scenarios as described in Sec. 3.2. We categorize our analysis into three cases: (a) intra-object context masking, where occlusions occur within the object of interest; (b) inter-object context masking, where occlusions occur within non-target objects; and (c) text context masking, where parts of textual descriptions are obscured.

In scenario (a), CARIS (Liu et al., 2023c) often struggles to recognize objects that are only partially visible, highlighting its difficulty in handling incomplete visual information. The challenge is not limited to occlusions of the target objects. In scenario (b), CARIS (Liu et al., 2023c) also faces challenges when occlusions occur in non-target areas surrounding the target object, reducing its ability to identify the intended objects correctly. Similar limitations are observed in the text description domain, as shown in Figure C(c). While CARIS can successfully recognize the target objects with complete text descriptions, its performance drops when only partial text information is available. In contrast, our MaskRIS demonstrates robustness across these occlusion scenarios by improving its comprehension and utilization of diverse contextual details in the image and text data.

Figure D shows more qualitative examples on the RefCOCO dataset. As shown in Figure D(a), our MaskRIS accurately captures the target objects without over- or under-capturing them. In Figure D(b), it demonstrates strong robustness to occlusion, effectively identifying partially visible objects. Furthermore, as highlighted in Figure D(c), MaskRIS successfully aligns target objects with the given text descriptions. These results demonstrate that our approach significantly improves the robustness of the model and its ability to utilize textual clues for accurate object identification.

We also provide failure cases in Figure D(d). When the referred object in an image or the given text description is ambiguous or difficult to recognize, our approach sometimes struggles to identify the correct target. For instance, in the first example of the top row, MaskRIS fails to recognize the drawing ("flowers") on the "bottle", instead capturing a different object. Another failure occurs when text descriptions include contrasting relative positions, such as "left" and "right", which confuse the model. Insufficient text descriptions also pose significant challenges. When text descriptions do not sufficiently characterize the target object such as "yellow" or "her" in the bottom row, MaskRIS captures the other objects as the target. These examples illustrate that while MaskRIS shows significant improvements in robustness and accuracy across various scenarios, it can still be limited in handling complex visual scenes or overly vague and ambiguous text descriptions.

**(a) Intra-object context masking**

*"person holding a snowboard"*     *"far left bowl"*

*"container with the sliced strawberries"*     *"elephant on left"*

Image     GT     CARIS     MaskRIS          Image     GT     CARIS     MaskRIS

**(b) Inter-object context masking**

*"top bed"*     *"giraffe in back"*

*"man on left"*     *"zebra in the back"*

Image     GT     CARIS     MaskRIS          Image     GT     CARIS     MaskRIS

**(c) Text context masking**

*"denim jacket girl"*     *"denim jacket [MASK]"*          *"mom holding child"*     *"[MASK] holding child"*

*"guy sitting"*     *"[MASK] sitting"*          *"guy in middle back striped shirt"*     *"guy in middle [MASK] striped shirt"*

Image     GT     CARIS     MaskRIS          Image     GT     CARIS     MaskRIS

Figure C: Qualitative examples under various occluded contexts on the RefCOCO dataset. Although CARIS (Liu et al., 2023c) tends to be inaccurate under occluded contexts, MaskRIS produces accurate predictions, demonstrating its robustness to occlusion and incomplete information. For text context masking, the word highlighted in red is masked.

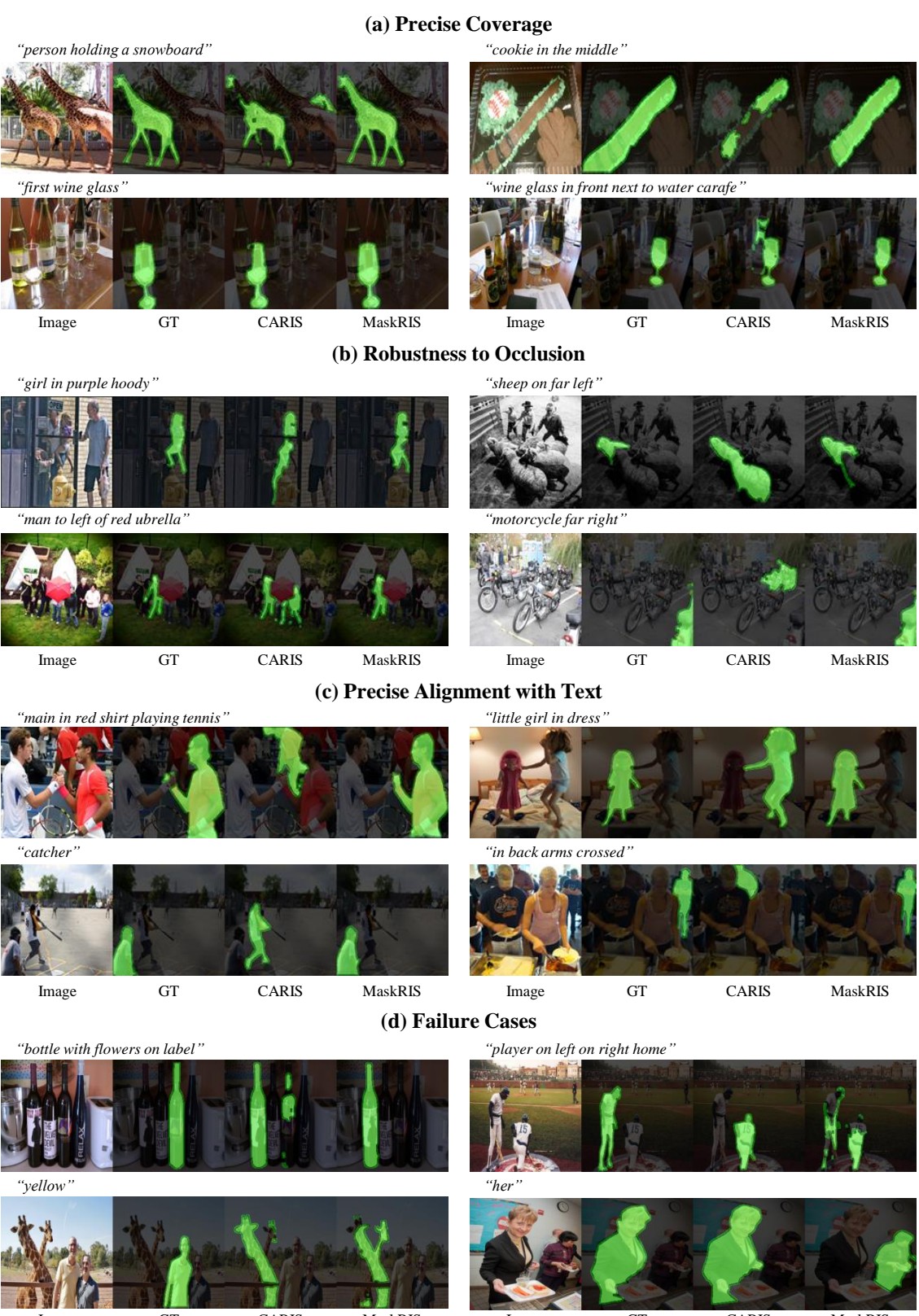

Figure D: More qualitative examples on the RefCOCO dataset. MaskRIS mitigates the limitations of CARIS (Liu et al., 2023c) by (a) capturing target objects more precisely, (b) being robust to occlusions, and (c) effectively using various textual clues. (d) We also provide failure cases.

## C  Implementation Details

Most of our experimental results are based on CARIS (Liu et al., 2023c). For the image encoder, we used the Swin-Base Transformer (Liu et al., 2021b), pre-trained on ImageNet-22k (Deng et al., 2009), and for the text encoder, we employed BERT-Base (Devlin et al., 2018). The maximum length of the text is set to 20 words. We used the AdamW (Loshchilov & Hutter, 2017) optimizer with a weight decay of 0.01. We applied different learning rates of $1e-5$ and $1e-4$ to encoders and the others, respectively, with a polynomial learning rate schedule with a power of 0.9. The model was trained for 50 epochs with a batch size of 16, and the input images were resized to $448 \times 448$.

To validate the adaptability of MaskRIS with other RIS methods, we integrated it into several existing models in both fully and weakly supervised settings, as shown in Table 2. Importantly, we follow the original training recipes of each method without any changes to the original architecture. This simple integration demonstrates the flexibility of MaskRIS and its seamless compatibility with existing frameworks.

