# OpenReview forum: "MaskRIS: Semantic Distortion-aware Data Augmentation for Referring Image Segmentation"
_TMLR — Accepted by TMLR_

### Review · Reviewer_2isH · 2025-07-20

**Summary Of Contributions:**

This paper introduces MaskRIS, a method addressing semantic distortion in data augmentation for Referring Image Segmentation (RIS), motivated similarly to CM-MaskSD [TMM 2024] by proposing an input masking strategy that simultaneously masks image patches and textual tokens, and a Distortion-aware Contextual Learning (DCL) module that employing self-distillation to enhance robustness against occlusions. The proposed method achieves SoTA performance on three RIS benchmarks.

**Audience:**

Yes

**Claims And Evidence:**

Yes

**Requested Changes:**

Major:

1. Clarify technological contribution. Clearly highlight the unique technological contributions of the proposed method and explicitly differentiate the DCL module and holistic masking strategy from existing masking RIS works (e.g., CM-MaskSD) in the introduction section.

2. Clarify experimental settings. Justify and clearly explain the choice of using reproduced results for CARIS under the "Standard" setting and reported results under the "Combined" setting, given the open-source availability and add pseudo code.

3. Expand comparisons to recent SoTA methods. Include results and integration studies with more recent RIS methods (e.g., MagNet, ReMamber, CM-MaskSD), and provide discussion on compatibility and improvements.

4. Prove MaskRIS’s generalizability. Evaluate the effectiveness of MaskRIS on a CLIP-based RIS models.

5. Extend ablation studies. Please refer to W.5.

Minor:

1. It is recommended to move Section A.1 from the Supplementary to main paper.

2. It is recommended to compare the different between MaskRIS and existing masking RIS works (e.g., CM-MaskSD) in your motivation figure (i.e., Figure 1)

3. It is recommended to compare the performance with CM-MaskSD on quantitative and qualitative experiments (i.e., Table1 and Figure 7)

**Strengths And Weaknesses:**

Strengths

1.	The idea of this paper is intuitive and easy to understand.

2.	Leverage masking strategy and self-distillation framework for RIS is reasonable.

3.	The framework is compatible with various RIS architectures, making it potentially useful as a plug-and-play method.

Weakness

1.  Limited technological contribution. The proposed MaskRIS is largely inspired by CM-MaskSD [1]. The CM-MaskSD already introduced a masked self-distillation method (i.e., Cross-Modality Masked Self-Distillation, Section III.B). However, the authors do not explicitly discuss the difference between their Distortion-aware Contextual Learning (DCL) module and the CM-MaskSD framework, making it unclear what the true contribution is. The main contribution appears to be replacing CM-MaskSD’s object-centric masking strategy with a holistic masking approach.  Therefore, I concern their might not meet the bar for TMLR and may not be of enough interest to the TMLR audience.

2. Unclear experiment setting. The authors report their own reproduced results for CARIS [2] under the "Standard" setting, while using the originally reported results for CARIS under the "Combined" setting. Given that the authors do not provide the source code and CARIS is open-source, the reason behind these choices should be clarified.

3. Missing comparison to recent SoTA methods. The comparison mainly includes methods from 2023, while overlooking newer RIS SoTA methods (e.g., [1, 4, 6]). It is also necessary to evaluate the integration of MaskRIS with recent SoTA RIS methods, such as MagNet [3] and ReMamber [6].

4. Concerns about generalizability. The authors claim that MaskRIS is a plug-and-play approach for various RIS models. However, Table 4 shows experiments mainly on text and image encoders that lack explicit alignment knowledge. There is insufficient evidence that the masking strategy is suitable for RIS models with cross-modal alignment knowledge (e.g., CLIP-based RIS [1,4]).

5. Need more ablation studies. For example, what is the performance of random patch masking? What is the performance of different word masking strategies? What is the performance of MaskRIS to different distillation strategies (e.g., using cross-modal contrastive loss)?  What is the performance of combine MaskRIS to recent SoTA methods (e.g., MagNet[3], ReMamber [6] or CM-MaskSD[1]).

References

[1] Wang, Wenxuan, et al. "Cm-masksd: Cross-modality masked self-distillation for referring image segmentation." IEEE Transactions on Multimedia 26 (2024): 6906-6916.

 [2] Sun-Ao Liu, Yiheng Zhang, Zhaofan Qiu, Hongtao Xie, Yongdong Zhang, and Ting Yao. Caris: Context aware referring image segmentation. In Proceedings of the 31st ACM International Conference on Multimedia, pp. 779–788, 2023c.

[3] Chng, Yong Xien, et al. "Mask grounding for referring image segmentation." Proceedings of the IEEE/CVF Conference on Computer Vision and Pattern Recognition. 2024.

[4] Huang, Jiaqi, et al. "Densely connected parameter-efficient tuning for referring image segmentation." Proceedings of the AAAI Conference on Artificial Intelligence. Vol. 39. No. 4. 2025.

[5] Kim, Seoyeon, et al. "Extending clip's image-text alignment to referring image segmentation." arXiv preprint arXiv:2306.08498 (2023).

[6] Yang, Yuhuan, et al. "Remamber: Referring image segmentation with mamba twister." European Conference on Computer Vision. Cham: Springer Nature Switzerland, 2024.

---

> ### Author Response · Authors · 2025-08-22
> **Official Comment by Authors (1/3)**
>
> We sincerely thank the reviewer for the valuable comments and constructive feedback. These insights have been very helpful in clarifying our contributions. We have revised the manuscript accordingly (revisions are highlighted in red) and made every effort to address each concern carefully, as detailed below.
>
> ---
> $$
> \mathbf{(W1)}\ \text{Clarify technological contribution.}
> $$
>
> We understand the reviewer’s concern that CM-MaskSD and our proposed MaskRIS may appear similar since both employ masking. We would like to clarify that, while both methods rely on masking as a tool, their novelty lies in how masking is adapted to distinct challenges in RIS—CM-MaskSD focuses on fine-grained alignment, whereas MaskRIS addresses semantic distortion introduced by augmentation. We appreciate this observation, as it allows us to more explicitly highlight the conceptual and technical differences between the two approaches.
>
> **On CM-MaskSD:**
>
> CM-MaskSD addresses mis-alignment issues, particularly over- and under-segmentation of target objects, which arise from insufficient fine-grained alignment between visual patches and text tokens. Its contribution lies in object-centric masking with self-distillation, leveraging the property of pre-aligned vision–language models that a global feature from one modality naturally correlates with patch or token features from the other. This correlation-guided masking strengthens fine-grained alignment and alleviates segmentation errors. Their masking strategy is specifically designed to enhance fine-grained alignment in RIS, with a particular focus on object regions through masking and self-distillation.
>
> **On MaskRIS:**
>
> In contrast, MaskRIS is motivated by a different and previously overlooked challenge: the semantic distortion caused by conventional data augmentation in RIS. To address this, we propose two key components:
> - **Holistic Context Masking (Image + Text)**: Unlike CM-MaskSD’s object-centric masking that focuses on specific regions or words, our holistic masking spans the entire image and referring expression—including both target and surrounding context—so the model learns to reason under incomplete or distorted inputs. This design aims  at a more general form of robustness, rather than being limited to target objects.
> - **Distortion-aware Contextual Learning (DCL)**: A dual-path distillation framework at the segmentation mask level, designed explicitly for robustness against semantic distortion, without architectural constraints or inference overhead.
>
> **Key Clarification:**
>
> Both CM-MaskSD and MaskRIS employ masking, but this does not diminish the novelty of either work—masking is a broadly used operation and not unique to any single method. If CM-MaskSD’s novelty lies in applying object-centric masking to improve patch–text alignment and reduce segmentation errors, then MaskRIS is equally novel in employing a fundamentally different strategy—holistic masking and distortion-aware distillation—to resolve augmentation-induced semantic distortion. Importantly, unlike CM-MaskSD, whose masking relies on correlations within pre-aligned vision–language representations, MaskRIS operates without such constraints, achieving a more general form of robustness that is broadly applicable and architecture-agnostic.
>
> To ensure this distinction is clearer to readers, we have revised the Introduction (Sec. 1), added a dedicated comparison figure (Figure 2) contrasting masking granularity and architectural dependency, and moved the supplementary discussion (previously Sec. A.1) into the main text (Sec. 4.4).
>
> In summary, MaskRIS’s novelty lies in (i) identifying and tackling the previously overlooked problem of augmentation-induced semantic distortion in RIS, and (ii) introducing a holistic context masking + distortion-aware dual-path distillation framework that is conceptually, operationally, and empirically distinct from prior masking-based RIS approaches. Furthermore, MaskRIS is architecture-agnostic and can be seamlessly extended to other vision–language tasks such as Referring Expression Comprehension (REC), as shown in Response to gCJq. Finally, we will release the full source code, which we believe will provide substantial value to the community by enabling transparent evaluation and easy adoption in diverse RIS and REC pipelines.

---

> ### Author Response · Authors · 2025-08-22
> **Official Comment by Authors (2/3)**
>
> $$
> \mathbf{(W2)}\ \text{Clarify experimental settings.}
> $$
>
> We sincerely apologize for the confusion regarding the experimental settings.
>
> For the CARIS “Combined” setting, we have now updated all reproduced results in Table 1 for consistency and clarity. In addition, we have included the full source code in the Appendix to ensure full transparency and reproducibility.
>
> | Method             | RefCOCO@val   | RefCOCO@testA | RefCOCO@testB |
> |--------------------|-------|-------|-------|
> | CARIS (paper)      | 76.63 | 79.40 | 73.52 |
> | CARIS (reproduced) | 76.67 | 79.85 | 73.07 |
> | MaskRIS            | 78.71 | 80.64 | 75.10 |
>
> Importantly, we would like to emphasize that all training configurations strictly follow the original CARIS implementation, including learning rate, optimizer, scheduler, and data preprocessing. The only modification is the addition of MaskRIS-specific masking ratio parameters, which are orthogonal to CARIS’s baseline setup and do not affect its original training pipeline.
>
> Notably, the reproduced results continue to show the same performance trend: MaskRIS consistently outperforms CARIS across all splits. This demonstrates that the reported improvements are robust and reproducible, regardless of whether original or reproduced results are used. We hope this resolves the confusion and reinforces the rigor of our experimental methodology.
>
> ---
>
> $$
> \mathbf{(W3,4)}\ \text{Expand comparisons to recent SoTA methods and prove MaskRIS’s generalizability. }
> $$
>
> We fully acknowledge the importance of including recent RIS baselines. However, MagNet and CM-MaskSD do not currently provide public training code, which made direct experimental integration infeasible at this stage.
>
> Instead, we carefully reproduced the experimental setups of ReMamber, a recently proposed RIS model, and evaluated the integration of MaskRIS into its pipeline. We observed a 1.65% improvement in oIoU (from 72.37 → 74.02) on the RefCOCO validation set, demonstrating that MaskRIS enhances segmentation accuracy. Importantly, ReMamber’s modular design allowed MaskRIS to plug in without additional architectural modifications, illustrating high compatibility. To further validate MaskRIS’s generalizability, we extended experiments to DETRIS, a recent CLIP-based RIS model. Integrating MaskRIS into DETRIS resulted in a 1.61% gain in oIoU (from 74.47 → 76.08) and consistent improvements across test splits (e.g., +1.06% on testA and +1.38% on testB). This confirms that MaskRIS is not only compatible with CLIP-based RIS pipelines but also provides consistent and measurable performance improvements.
>
> Notably, Table 2 already demonstrates MaskRIS’s strong compatibility with other CLIP-based RIS models such as ETRIS and CRIS, both of which show measurable gains when MaskRIS is applied. This highlights MaskRIS’s plug-and-play design across diverse CLIP-based frameworks.
>
> | Method             | RefCOCO@val           | RefCOCO@testA         | RefCOCO@testB         |
> |--------------------|---------------|---------------|---------------|
> | ReMamber           | 72.37         | 74.85         | 68.29         |
> | ReMamber + MaskRIS | 74.02 (+1.65) | 76.32 (+1.47) | 69.06(+0.77)  |
> | DETRIS             | 74.47         | 77.40         | 71.46         |
> | DETRIS + MaskRIS   | 76.08 (+1.61) | 78.46 (+1.06) | 72.84 (+1.38) |

---

> ### Author Response · Authors · 2025-08-22
> **Official Comment by Authors (3/3)**
>
> $$
> \mathbf{(W5)}\ \text{Extend ablation studies. }
> $$
>
> We thank the reviewer for the suggestion to expand our ablation experiments. We have now conducted further studies to isolate the contributions of individual components.
>
> **(1) Patch masking strategies.** As shown in Supp. Figure A and Table C, we compared four sampling approaches—patch-wise, grid-wise, block-wise, and Cutout—under the same masking ratio (75%). Patch-wise random masking achieved the highest performance (76.49 oIoU on RefCOCO val), confirming that finer-grained, stochastic masking improves data diversity and reduces distortion compared to structured or large-block removal. All variants still surpass CARIS (74.67 oIoU), validating the robustness of our masking design.
>
> |      | Patch-wise | Grid-wise | Block-wise | Cutout |
> |------|------------|-----------|------------|--------|
> | RefCOCO@val | 76.49      | 76.14     | 75.90      | 76.07  |
>
> **(2) Word masking strategies.** To analyze linguistic perturbation, we compared four text masking schemes: token-level random masking (baseline), Whole-Word Masking (WWM), SpanBERT-style span masking, and a POS-aware strategy. As reported in Supp. Table D, the baseline MaskRIS achieves 76.49 oIoU on RefCOCO val. WWM slightly reduces performance (76.03), likely because masking entire words often removes key entities, while span masking performs comparably to the baseline (76.46). The POS-aware strategy, which preserves entity-bearing nouns and relational terms while randomly masking adjectives and adverbs, achieves the best result (76.52). Notably, all masking variants still surpass the previous SoTA CARIS (74.67 oIoU), highlighting the robustness of our framework regardless of the specific masking choice.
>
> |      | MaskRIS | WWM   | SpanBERT | POS-based |
> |------|---------|-------|----------|-----------|
> | RefCOCO@val | 76.49   | 76.03 | 76.46    | 76.52     |
>
> **(3) Cross-distillation objectives.** We further investigated whether incorporating a cross-modal contrastive loss (CM-MaskSD) could enhance our distortion-aware contextual learning (DCL) framework. As detailed in Supp. TableE and SectionB.3, adding cross-distillation yields a slight gain on the validation set (76.68 vs. 76.49), but this improvement does not generalize to the test splits. On RefCOCO testA, performance remains nearly unchanged (78.92 vs. 78.96), and on testB it even decreases (73.21 vs. 73.96). Moreover, this setting substantially increases computational cost, as each training step requires three forward passes instead of one. These results confirm that cross-distillation provides no consistent accuracy benefit and leads to an unfavorable trade-off between performance and efficiency.
>
> | Method             | RefCOCO@val   | RefCOCO@testA | RefCOCO@testB |
> |--------------------|-------|-------|-------|
> | DCL (Ours)         | 76.49 | 78.96 | 73.96 |
> | Cross-distillation | 76.68 | 78.92 | 73.21 |

---

> > ### Comment · Action_Editor_K9ka · 2025-10-12
> > **Please enter official recommendation**
> >
> > Dear reviewer 2isH,
> >
> > This is a gentle reminder regarding the manuscript you reviewed. The authors have submitted their response to the reviews. It would be grateful if you could review their feedback and provide your final recommendation at your earliest convenience.
> >
> > Your AE.

---

> > > ### Comment · Reviewer_2isH · 2025-10-14
> > >
> > > After considering the rebuttal, I find the paper clearly written and more valuable.
> > >
> > > While the contribution over CM-MaskSD is incremental, the authors
> > >
> > > 1. explicitly target augmentation-induced semantic distortion.
> > > 2. propose a holistic image-and-text masking scheme with a distortion-aware distillation objective.
> > > 3.  demonstrate compatibility across architectures, including improvements when plugging MaskRIS into recent pipelines (e.g., ReMamber) and a CLIP-based RIS model (e.g., DETRIS).
> > >
> > > I believe the method’s plug-and-play nature makes it a useful addition to the RIS toolkit.

---

### Review · Reviewer_gCJq · 2025-07-25

**Summary Of Contributions:**

The paper proposes a masking-based data augmentation method to assist with referring image segmentation training. Unlike traditional methods, which could potentially distort the image and cause a performance drop, the proposed method applies random masks to both image and text, enabling a performance boost.

**Audience:**

Yes

**Claims And Evidence:**

Yes

**Requested Changes:**

* Add performance on REC tasks and some analysis

**Strengths And Weaknesses:**

Strength
* The paper addresses a practical question of referring image segmentation: augmentation can distort the object in the image that the text is referring to. Solving this problem is meaningful for RIS tasks.
* The masking mechanism is simple and effective
* The ablation experiment is comprehensive.

Weakness/Questions
* The data augmentation dilemma seems to be universal to the REC and RIS task? How would the same augmentation technique apply to REC tasks?
* The distortion-aware loss seems to give marginal improvement while requiring twice the inference time during training. Do you have some cost vs performance analysis on this part?
* Would this method be applicable to those VLM-based generalists like CogVLM-Grounding?

---

> ### Author Response · Authors · 2025-08-22
> **Official Comment by Authors (1/2)**
>
> We sincerely thank the reviewer for the valuable comments and constructive feedback. These insights have been very helpful in clarifying our contributions. We have revised the manuscript accordingly (revisions are highlighted in red) and made every effort to address each concern carefully, as detailed below.
>
> ---
> $$
> \mathbf{(W1)}\ \text{How would the same augmentation technique apply to REC tasks?}
> $$
>
> | Method             | RefCOCO@val   | RefCOCO@testA | RefCOCO@testB |
> |--------------------|---------------|---------------|---------------|
> | EEVG (ECCV’24)     | 87.87         | 89.97         | 85.34         |
> | EEVG + MaskRIS     | 88.72 (+0.85) | 90.81 (+0.84) | 85.36 (+0.02) |
> | SimVG (NeurIPS’24) | 86.94         | 88.97         | 82.87         |
> | SimVG + MaskRIS    | 87.79 (+0.85) | 90.02 (+1.05) | 84.88 (+2.01) |
> | C3VG (AAAI’25)     | 92.42         | 94.12         | 89.34         |
> | C3VG + MaskRIS     | 93.29 (+0.87) | 94.86 (+0.74) | 90.29 (+0.95) |
>
> We appreciate the reviewer’s insightful question regarding whether the data augmentation dilemma observed in RIS (Referring Image Segmentation) extends to REC (Referring Expression Comprehension), and how MaskRIS can be applied in that context.
>
> **On the universality of the augmentation dilemma.** The conflict between conventional image augmentations (e.g., flipping, cropping, color jitter) and language-grounded tasks is not unique to RIS. REC also relies on precise spatial and attribute alignment between text and image. For example, flipping an image containing the phrase “the man on the left” alters the semantic meaning for both RIS and REC. This confirms that the augmentation dilemma is indeed universal to referring tasks.
>
> **Applying MaskRIS to REC.** To investigate this, we integrated MaskRIS into various REC models (EEVG [1], SimVG [2], C3VG [3]). The above table (Table 3 in manuscript) show that MaskRIS benefits REC in the same way it benefits RIS, by enhancing robustness to occlusion and incomplete linguistic cues while avoiding augmentation-induced label inconsistencies. The findings confirm that MaskRIS is not task-specific. Its masking approach addresses the augmentation dilemma across both RIS and REC, making it a general training strategy for referring tasks where traditional augmentations are unsafe.
>
> ---
>
> $$
> \mathbf{(W2)}\ \text{Analysis of cost vs. performance for distortion-aware loss.}
> $$
>
> We thank the reviewer for pointing out the trade-off between performance gains and computational cost. In summary, the distortion-aware consistency loss improves RefCOCO validation by +0.78 oIoU, a substantial gain beyond marginal variation, while adding cost only during training—inference remains unchanged. When training efficiency is critical, alternatives such as masking alone or shorter training schedules still show clear improvements over the baseline. The detailed analysis is as follows.
>
> **Cost vs. Performance Evidence.** As shown in Table 4, adding DCL yields an additional **+0.78 oIoU on RefCOCO validation (75.71 → 76.49)** when combined with our masking strategies. This confirms that DCL refines the model’s robustness by aligning predictions between original and masked inputs, complementing the masking strategy.
>
> The table below (Appendix Table H) presents a more detailed cost analysis. Because DCL introduces a dual-path during training, the forward pass count doubles. However, MaskRIS trained for just 25 epochs already surpasses CARIS at 50 epochs (75.87 vs. 74.67 oIoU). With the standard 50 epochs, MaskRIS achieves 76.49 oIoU, even outperforming CARIS trained for twice as long (100 epochs). This indicates that the gains are not a mere byproduct of longer training or extra compute, but rather from DCL’s consistency loss guiding the network to leverage masked inputs effectively.
>
> | Method  | Epoch | RefCOCO@val   | RefCOCO@testA | RefCOCO@testB |
> |---------|-------|-------|-------|-------|
> | CARIS   | 50    | 74.67 | 77.63 | 71.71 |
> | CARIS   | 100   | 74.80 | 77.38 | 70.63 |
> | MaskRIS | 25    | 75.87 | 77.99 | 73.51 |
> | MaskRIS | 50    | 76.49 | 78.96 | 73.96 |
>
> **Training vs. Inference Cost.** Importantly, DCL only affects training. The inference pipeline remains unchanged—MaskRIS introduces no architectural modifications and no extra inference-time cost. The one-time cost increase during training is therefore amortized, especially in scenarios where the model is deployed broadly.
>
> **Cost-sensitive alternatives.** If training cost is a critical constraint, Table 4 shows that our masking strategy alone (without DCL) still yields substantial performance gains over the baseline (75.71 oIoU vs. 74.67). Likewise, as demonstrated in Table H, even when the training epochs are reduced (e.g., 25 epochs), MaskRIS continues to outperform the CARIS baseline trained for longer. This means practitioners can either drop DCL or shorten training schedules and still achieve meaningful improvements without incurring the full computational overhead.

---

> ### Author Response · Authors · 2025-08-22
> **Official Comment by Authors (2/2)**
>
> $$
> \mathbf{(W3)}\ \text{Would this method be applicable to those VLM-based generalists like CogVLM-Grounding? }
> $$
>
> Yes, MaskRIS can be applied to VLM-based generalist models such as CogVLM-Grounding.
>
> **Compatibility with VLM-based generalists.** MaskRIS is designed as a training-time data augmentation and regularization strategy. It does not require any architectural modifications to the base model. Because of this plug‑and‑play nature, it can be integrated into generalist VLMs (e.g., CogVLM-Grounding) simply by introducing image masking + text masking during fine-tuning or task-specific adaptation.
>
> Even for powerful VLM-based generalists, conventional augmentations (flip, crop, color jitter) can contradict language cues (e.g., “object on the right”) or break spatial grounding. MaskRIS, by contrast, preserves the semantic integrity of image–text pairs while still expanding training diversity. This property makes it a safe and effective augmentation choice for VLM fine-tuning. Although we have not yet run full experiments on CogVLM-Grounding, our successful integrations with RIS and REC models demonstrate that MaskRIS Integrates seamlessly without altering the model backbone and Improves grounding performance.
>
> ---
> $$
> \mathbf{References}
> $$
>
> [1] Chen et al., An efficient and effective transformer decoder-based framework for multi-task visual grounding, ECCV 2024
> [2] Dai et al., Simvg: A simple framework for visual grounding with decoupled multi-modal fusion, NeurIPS 2024
> [3] Dai et al., Multi-task visual grounding with coarse-to-fine consistency constraints, AAAI 2025

---

### Review · Reviewer_3PtX · 2025-08-14

**Summary Of Contributions:**

This paper first analysis the limitation of existing data augmentation methods for referring image segmentation. It then proposes MaskRIS, a framework aiming to improve robustness for referring image segmentation via image masking, text masking, and Distortion-aware Contextual Learning (DCL). Experimental results demonstrates the method's effectiveness on the RefCOCO series benchmarks.

**Audience:**

Yes

**Broader Impact Concerns:**

Not much.

**Claims And Evidence:**

Yes

**Requested Changes:**

Please refer to the weaknesses section.

**Strengths And Weaknesses:**

### Strengths

- The paper is overall clear and easy to follow.
- MaskRIS is conceptually simple, does not require architectural changes, and is applicable to a range of existing RIS models.
- Experimental results across RefCOCO/g/+ benchmarks and different model structures demonstrate the effectiveness of the MaskRIS.

### Weaknesses
- Dataset scope is somewhat limited. All images are from MS-COCO datasets. Could the authors experiment with  datasets from other image sources?
- Certain implementation details are missing. E.g. What is the ratio of the CE loss and the distillation loss? Does this ratio affect the results?
- Ablations could be more comprehensive. E.g. It would be valuable to assess the sensitivity of performance to the choice of patch size in image masking.
- The evaluation would be more convincing if it included additional qualitative and quantitative results demonstrating MaskRIS’s robustness under both image and text distortions.

---

> ### Author Response · Authors · 2025-08-22
> **Official Comment by Authors**
>
> We sincerely thank the reviewer for the valuable comments and constructive feedback. These insights have been very helpful in clarifying our contributions. We have revised the manuscript accordingly (revisions are highlighted in red) and made every effort to address each concern carefully, as detailed below.
>
> ---
> $$
> \mathbf{(W1)}\ \text{Generality beyond MS-COCO datasets.}
> $$
>
>
> In addition to the MS COCO–based benchmarks, we also validated MaskRIS on RefClef, a subset of the ImageCLEF dataset. Unlike MS COCO, RefClef features more diverse natural scenes and covers a broader set of object categories that are not restricted to MS-COCO’s predefined classes. As shown in Appendix Table A, MaskRIS consistently improves performance over the baseline (76.06 vs. 75.21), confirming that the effectiveness of our approach is not limited to COCO-style data.
>
> We thank the reviewer for raising this concern. Following the common practice of prior works in RIS, our main experiments are conducted on MS COCO–based benchmarks. Still, we agree that validation beyond COCO is important, and our additional results on RefClef demonstrate that the proposed holistic masking and distortion-aware training framework generalizes well across datasets with different scene compositions and category coverage, thereby reinforcing the broader applicability of MaskRIS.
>
>
> | Method          | Image Encoder | Text Encoder | oIoU @ RefClef  |
> |-----------------|---------------|--------------|-----------------|
> | CARIS           | Swin-B        | BERT         | 75.21           |
> | CARIS + MaskRIS | Swin-B        | BERT         | 76.06 (+0.85)   |
>
> ---
>
> $$
> \mathbf{(W2)}\ \text{What is the ratio of CE loss to distillation loss, and how does it affect the results?}
> $$
>
> We reported the ratio between the CE loss and the distillation loss (λ = 0.5 by default) in Appendix Table F. But, as the reviewer pointed out, this detail was not sufficiently visible in the main text. We have therefore updated Sec. 3.3 to explicitly mention the weighting factor and to direct readers to the Appendix, which further shows that MaskRIS is robust across a wide range of λ values (with λ = 0.25 giving the best accuracy and λ = 0.5 still yielding strong performance).
>
> ---
> $$
> \mathbf{(W3)}\ \text{What is the impact of varying patch sizes on performance in image masking?}
> $$
>
> We agree that analyzing patch size is important. In Appendix Table E, we provide ablations over multiple patch sizes (8, 16, 32, 64, 112) for image masking. The results show consistently strong performance across different patch sizes, with only marginal variation. This demonstrates that MaskRIS is stable and not overly sensitive to this design choice.
>
> | patch size | 8     | 16    | 32    | 64    | 112   |
> |------------|-------|-------|-------|-------|-------|
> | RefCOCO@val       | 76.58 | 76.49 | 76.49 | 76.47 | 76.01 |
>
> ---
> $$
> \mathbf{(W4)}\ \text{Additional qualitative and quantitative results on robustness to image/text distortions.}
> $$
>
> We also provide extensive qualitative and quantitative evidence of robustness under both image and text distortions. Appendix Figure C illustrates MaskRIS’s performance under intra-object, inter-object, and text-level occlusions, while Appendix Figure D shows further examples highlighting precise coverage, robustness to occlusion, and alignment with incomplete text. In addition, Figure 7 in the main paper presents side-by-side comparisons under various distortion scenarios, further confirming that MaskRIS consistently outperforms the baseline. These results together demonstrate that MaskRIS is significantly more resilient to semantic distortions than prior approaches.

---

### Decision · Action_Editor_K9ka · 2025-10-23

**Recommendation:** Accept as is

**Audience:**

Yes

**Audience Explanation:**

This paper tackles the fundamental problem in computer vision, demonstrating clear improvements and robustness. AC confirms that the paper will interest many readers in this field.

**Claims And Evidence:**

Yes

**Claims Explanation:**

This paper introduces MaskRIS for the referring image segmentation (RIS) problem. Instead of conventional augmentation techniques, it randomly masks parts of the image and the text. The approach employs Distortion-aware Contextual Learning to train the model to make correct predictions even with incomplete information.

The concerns in the initial review were about (1) an existing work (CM-MaskSD) proposed a similar idea, (2) unclear aspects in the experimental settings, (3) comparison with the latest approach, and (4) generalizability when adopted for other architectures. After the rebuttal, the reviewer successfully resolved these concerns, and all reviewers stated 'Learning Accept' scores.

AE confirms that the joint image-text masking scheme, combined with the distillation objective, is effective, and AE values the ability of this idea to be adopted for recent RIS pipelines. Since the major concerns have been properly addressed, AE recommends 'acceptance as is' for this paper.